# A RELATIONAL INTERVENTION APPROACH FOR UNSUPERVISED DYNAMICS GENERALIZATION IN MODEL-BASED REINFORCEMENT LEARNING

**Jiaxian Guo**[†][*]   **Mingming Gong**[‡]   **Dacheng Tao**[†][§]
The University of Sydney[†]   The University of Melbourne[‡]   JD Explore Academy[§]
`jguo5934@uni.sydney.edu.au`   `mingming.gong@unimelb.edu.au`
`dacheng.tao@gmail.com`

## ABSTRACT

The generalization of model-based reinforcement learning (MBRL) methods to environments with unseen transition dynamics is an important yet challenging problem. Existing methods try to extract environment-specified information $Z$ from past transition segments to make the dynamics prediction model generalizable to different dynamics. However, because environments are not labelled, the extracted information inevitably contains redundant information unrelated to the dynamics in transition segments and thus fails to maintain a crucial property of $Z$: $Z$ should be similar in the same environment and dissimilar in different ones. As a result, the learned dynamics prediction function will deviate from the true one, which undermines the generalization ability. To tackle this problem, we introduce an interventional prediction module to estimate the probability of two estimated $\hat{z}_i, \hat{z}_j$ belonging to the same environment. Furthermore, by utilizing the $Z$'s invariance within a single environment, a relational head is proposed to enforce the similarity between $\hat{Z}$ from the same environment. As a result, the redundant information will be reduced in $\hat{Z}$. We empirically show that $\hat{Z}$ estimated by our method enjoy less redundant information than previous methods, and such $\hat{Z}$ can significantly reduce dynamics prediction errors and improve the performance of model-based RL methods on zero-shot new environments with unseen dynamics. The codes of this method are available at `https://github.com/CR-Gjx/RIA`.

## 1   INTRODUCTION

Reinforcement learning (RL) has shown great success in solving sequential decision-making problems, such as board games (Silver et al., 2016; 2017; Schrittwieser et al., 2020), computer games (*e.g.* Atari, StarCraft II) (Mnih et al., 2013; Silver et al., 2018; Vinyals et al., 2019), and robotics (Levine & Abbeel, 2014; Bousmalis et al., 2018). However, solving real-world problems with RL is still a challenging problem because the sample efficiency of RL is low while the data in many applications is limited or expensive to obtain (Gottesman et al., 2018; Lu et al., 2018; 2020; Kiran et al., 2020). Therefore, model-based reinforcement learning (MBRL) (Janner et al., 2019; Kaiser et al., 2019; Schrittwieser et al., 2020; Zhang et al., 2019b; van Hasselt et al., 2019; Hafner et al., 2019b;a; Lenz et al., 2015), which explicitly builds a predictive model to generate samples for learning RL policy, has been widely applied to a variety of limited data sequential decision-making problems.

However, the performance of MBRL methods highly relies on the prediction accuracy of the learned environmental model (Janner et al., 2019). Therefore, a slight change of environmental dynamics may cause a significant performance decline of MBRL methods (Lee et al., 2020; Nagabandi et al., 2018a; Seo et al., 2020). The vulnerability of MBRL to the change of environmental dynamics makes them unreliable in real world applications. Taking the robotic control as an example (Nagabandi et al., 2018a; Yang et al., 2020; Rakelly et al., 2019; Gu et al., 2017; Bousmalis et al., 2018; Raileanu et al., 2020; Yang et al., 2019), dynamics change caused by parts damages could easily lead to the failure of MBRL algorithms. This problem is called the *dynamics generalization* problem in MBRL, where the training environments and test environments share the same state $\mathcal{S}$ and action space $\mathcal{A}$ but the transition dynamics between states $p(s_{t+1}|s_t, a_t)$ varies across different environments. Following

previous works (Petersen et al., 2018; Gottesman et al., 2018), we focus on the **unsupervised dynamics generalization** setting, *i.e.* the id or label information of dynamics function in training MDPs is not available. This setting appears in a wide range of applications where the information of dynamics function is difficult to obtain. For example, in healthcare, patients may respond differently to the same treatment, *i.e.*, $p(s_{t+1}|s_t, a_t)$ varies across patients. However, it is difficult to label which patients share similar dynamics.

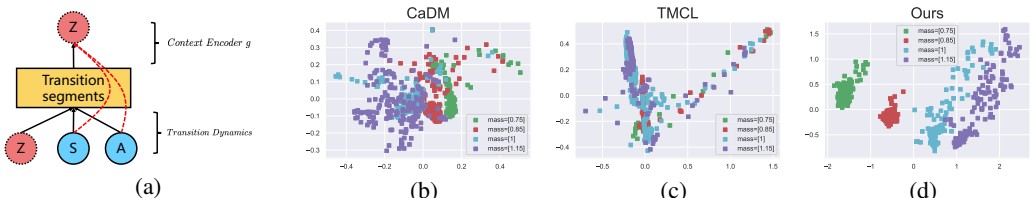

(a)  (b)  (c)  (d)

Figure 1: (a) The illustration of why historical states and actions are encoded in environment-specified factor $Z$, (b)(c)(d) The PCA visualizations of estimated context (environmental-specific) vectors in **Pendulum** task, where the dots with different colors denote the context vector (after PCA) estimated from different environments. More visualization results are given at Appendix A.13.

To build a generalized dynamics prediction function that can generalize to different transition dynamics, the shift of the transition dynamics can be modelled as the change of unobserved factors across different environments, *i.e.* there are hidden environment-specified factors $Z \in \mathcal{Z}$ which can affect the environmental dynamics. This is analogous to the human intuition to understand the change of dynamics, *e.g.* patients may show different responses to identical treatments because the differences of their gene sequences can affect how well they absorb drugs (Wilke et al., 2007). It is natural to assume that $Z$ is the same in a single environment but varies across different environments. As such, these unobserved environment-specified factors do not violate the nature of MDP in a single environment, but their changes can affect the dynamics functions across environments. Therefore, the dynamics function $f : \mathcal{S} \times \mathcal{A} \to \mathcal{S}$ can naturally be augmented by incorporating $Z$ to be $f : \mathcal{S} \times \mathcal{A} \times \mathcal{Z} \to \mathcal{S}$ (Rakelly et al., 2019; Zhou et al., 2019; Lee et al., 2020; Seo et al., 2020).

Learning the augmented dynamics function is difficult because the environment-specified factor $Z$ is unobservable. Previous methods (Rakelly et al., 2019; Zhou et al., 2019; Lee et al., 2020) try to extract information from historical transition segments and use it as a surrogate for $Z$ (Figure 1a) . However, in the unsupervised dynamics generalization setting, the extracted information from historical transition segments inevitably contains redundant information unrelated to the dynamics. The redundant information would cause the surrogate for $Z$ to lose a crucial property that characterizes $Z$: $Z$ should be similar in the same environment and dissimilar in different environments. As shown in Figure 1b, the environment-specified information $\hat{Z}$ learned by CaDM (Lee et al., 2020) does not form clear clusters for different environments. Because the learned $\hat{Z}$ fails to represent environmental information, the learned dynamics function will deviate from the true one, which undermines the generalization ability. To alleviate this problem, TMCL (Seo et al., 2020) directly clusters the environments by introducing multiple prediction heads, *i.e.* multiple prediction functions. However, TMCL needs to choose the proper prediction head for each new environment, making it hard to be deployed into the scenario with consistently changing environments, *e.g.* robots walk in the terrain which is constantly changing. To avoid adaptation at the deployment time, we thus need to learn a single generalized prediction function $\hat{f}$. To ensure that $\hat{f}$ can learn modals of transition dynamics in different environments, we need to cluster $Z$ according to their belonging environments.

In this paper, we provide an explicit and interpretable description to learn $Z$ as a vector $\hat{Z}$ (*i.e.* the estimation of $Z$) from the history transition segments. To cluster $\hat{Z}$ from the same environment, we introduce a relational head module as a learnable function to enforce the similarity between $\hat{Z}$s learned from the same environments. However, because environment label is not available, we can only cluster the $\hat{Z}$s from the same trajectory, so we then propose an interventional prediction module to identify the probability of a pair of $\hat{z}_i, \hat{z}_j$ belonging to the same environment through estimating $\hat{Z}$'s direct causal effect on next states prediction by do-calculus (Pearl, 2000). Because $Z$s from the same environment surely have the same causal effect, we can directly maximize the similarity of $\hat{Z}$s with the similar causal effect using the relational head, and thus can cluster $\hat{Z}$ according to the estimated environmental similarity and alleviate the redundant information that varies in an environment, *e.g.* historical states and actions. In the experiments, we evaluate our method on a

range of tasks in OpenAI gym (Brockman et al., 2016) and Mujoco (Todorov et al., 2012), and empirically show that $\hat{Z}$ estimated by our method enjoy less redundant information than baselines. The experimental results show that our method significantly reduces the model prediction errors and outperforms the state-of-art model-based RL methods **without any adaptation step** on a new environment, and even achieve comparable results with the method directly cluster $\hat{Z}$ with the true environment label.

## 2 RELATED WORK

**Dynamics Generalization in MBRL**     Several meta-learning-based MBRL methods are proposed Nagabandi et al. (2018a;b); Sæmundsson et al. (2018); Huang et al. (2021) to adapt the MBRL into environments with unseen dynamics by updating model parameters via a small number of gradient updates Finn et al. (2017) or hidden representations of a recurrent model Doshi-Velez & Konidaris (2016), and then Wang & van Hoof (2021) proposes a graph-structured model to improve dynamics forecasting. Ball et al. (2021) focuses on the offline setting and proposes an augmented model method to achieve zero-shot generalization. Lee et al. (2020); Seo et al. (2020) try to learn a generalized dynamics model by incorporating context information or clustering dynamics implicitly using multi-choice learning, aiming to adapt any dynamics without training. However, how to explicitly learn the meaningful dynamics change information remains a big challenge.

**Relational Learning**     Reasoning relations between different entities is an important way to build knowledge of this world for intelligent agents Kemp & Tenenbaum (2008). In the past decades, relational paradigm have been applied to a wide range of deep learning-based application, *e.g.*, reinforcement learning Zambaldi et al. (2018), question-answer Santoro et al. (2017); Raposo et al. (2017), graph neural network Battaglia et al. (2018), sequential streams Santoro et al. (2018), few-shot learning Sung et al. (2018), object detection Hu et al. (2018) and self-supervised learning Patacchiola & Storkey (2020). Different from previous methods that perform binary relational reasoning on entities, our method can also perform multiplies relations between entities through the learned similarity of entities, and thus can learn more compact and meaningful entity representation.

**Causality in Reinforcement Learning**     Many works focus on the intersection area of reinforcement learning and causal inference. For example, some works aims to alleviate the causal confusion problem in the imitation learning (de Haan et al., 2019; Zhang et al., 2020c; Kumor et al., 2021), batch learning (Bannon et al., 2020), and partial observability settings (Forney et al., 2017; Kallus & Zhou, 2018; Zhang et al., 2019a; Lu et al., 2018) in the online environment (Lyle et al., 2021; Zhang & Bareinboim, 2018), (Wang et al., 2020a) also try to apply causal inference in the offline setting, where the observational data is always confounded. Lee & Bareinboim (2018); Bareinboim et al. (2015); Lattimore et al. (2016); Mozifian et al. (2020); Volodin et al. (2020) also explore how to design an optimal intervention policy in bandits or RL settings. In addition, (Zhang et al., 2020a;b) improve the generalization ability of state abstraction. Different from these methods, we focus on the setting of unsupervised dynamics generalization, and measure the direct causal effect (Pearl, 2013) between $\hat{Z}$ and the next state to estimate the probability of them belonging to the same environment.

## 3 METHODS

In the section, we first introduce the formulation of the dynamic generalization problem. Then we present the details on how relational intervention approach learns the environment-specified factors.

### 3.1 PROBLEM SETUP

The standard reinforcement learning task can be formalized as a Markov decision process (MDP) $\mathcal{M} = (\mathcal{S}, \mathcal{A}, r, p, \gamma, \rho_0)$ over discrete time (Puterman, 2014; Sutton & Barto, 2018), where $\mathcal{S}$, $\mathcal{A}$, $\gamma \in (0, 1]$, $\rho_0$ are state space, action space, the reward discount factor, and the initial state distribution, respectively. The reward function $r : \mathcal{S} \times \mathcal{A} \to \mathbb{R}$ specifies the reward at each timestep $t$ given $s_t$ and $a_t$, and transition dynamics $p(s_{t+1}|s_t, a_t)$ gives the next state distribution conditioned on the current state $s_t$ and action $a_t$. The goal of RL is to learn a policy $\pi(\cdot|s)$ mapping from state $s \in \mathcal{S}$ over the action distribution to maximize the cumulative expected return over timesteps $\mathbb{E}_{s_t \in \mathcal{S}, a_t \in \mathcal{A}}[\sum_{t=0}^{\infty} \gamma^t r(s_t, a_t)]$. In model-based RL, a model $f$ is used to approximate the transition dynamics $p$, and then $f$ can provide training data to train policy $\pi$ or predict the future sequences for

planning. Benefiting from data provided by learned dynamics model $f$, model-based RL has higher data efficiency and better planing ability compared with model-free RL.

Here we consider the unsupervised *dynamics generalization* problem in model-based RL, where we are given K training MDPs $\{\mathcal{M}_i^{tr}\}_{i=1}^K$ and L test MDPs $\{\mathcal{M}_j^{te}\}_{j=1}^L$ that have the same state and action space but disjoint dynamics functions, and we randomly sample several MDPs from training MDPs in each training iteration. We assume that all these MDPs have a finite number of dynamics functions, meaning that the MDPs can be categorized into a finite number of environments and the MDPs in each environment share the same dynamics function but the environment id of MDPs is unavailable in the training process. In the context of model-based RL, how to learn a generalized dynamics model $f$ is the key challenge to solve unsupervised *dynamics generalization* problem.

## 3.2 OVERVIEW

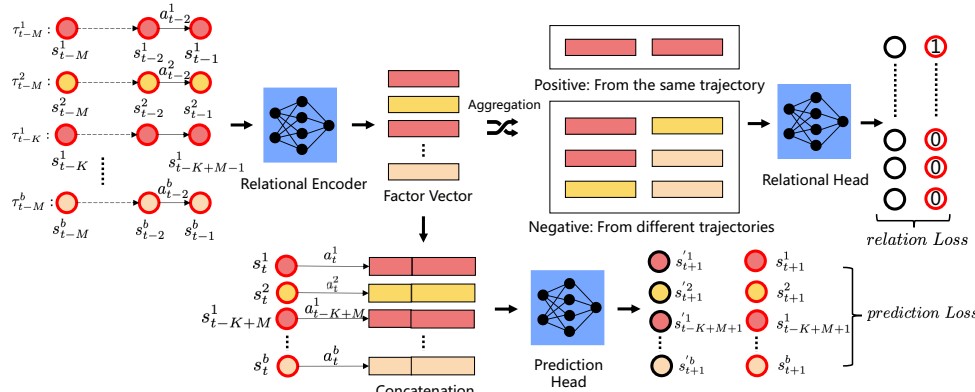

Figure 2: An overview of our Relational Intervention approach, where Relational Encoder, Prediction Head and Relational Head are three learnable functions, and the circles denote states (Ground-Truths are with red boundary, and estimated states are with black boundary), and the rectangles denote the estimated vectors. Specifically, *prediction Loss* enables the estimated environmental-specified factor can help the Prediction head to predict the next states, and the *relation Loss* aims to enforce the similarity between factors estimated from the same trajectory or environments.

As analyzed in Section 1, we can incorporate the environment-specified factors $Z \in \mathcal{Z}$ into the dynamics prediction process to generalize the dynamic functions on different environments, *i.e.* extending the dynamics function from $f : \mathcal{S} \times \mathcal{A} \to \mathcal{S}$ to $f : \mathcal{S} \times \mathcal{A} \times \mathcal{Z} \to \mathcal{S}$. Because $Z$ is the same within an environment, we expect estimated $\hat{Z}$s from the same environment are similar while those from different environments are dissimilar. Therefore, $f$ models the commonalities of the transition dynamics in different environments and $Z$ models the differences. In the supervised dynamics generalization setting, where the environment id is given, one can easily learn $Z$ by using metric losses, *e.g.*, CPC (Oord et al., 2018) and relation loss (Patacchiola & Storkey, 2020) to enforce that the estimated $\hat{Z}$s are similar in the same environment and dissimilar in different environments. However, since the environment label is unavailable in the unsupervised setting, we have to simultaneously learn $Z$ and discover the cluster structures. To this end, we propose an intervention module to measure the similarities between each pair of $\hat{z}_i$ and $\hat{z}_j$ as the probability of them belonging to the same environment. Furthermore, we then introduce a relational head to aggregate $\hat{Z}$s with high probability using relation loss. By simultaneously updating the dynamics prediction and the relation loss, we can cluster $\hat{Z}$s from the same environment, and learn an augmented dynamics prediction model $\hat{f}$. Next, we will give details about our relational intervention approach.

## 3.3 RELATIONAL CONTEXT ENCODER

To learn the environment-specified factor $Z$ of each environment, we firstly introduce a relational encoder $g$ parameterized by $\phi$. Similar to previous methods (Nagabandi et al., 2018a; Rakelly et al., 2019; Zhou et al., 2019; Lee et al., 2020; Seo et al., 2020), we use the past transition segments $\tau_{t-k:t-1} = \{(s_{t-k}, a_{t-k}), ..., (s_{t-1}, a_{t-1})\}$ as the input of $g$ to estimate its corresponding $\hat{z}_{t-k:t-1}$:

$$\hat{z}_{t-k:t-1} = g(\tau_{t-k:t-1}^i; \phi).$$

After obtaining environment-specified $\hat{z}_{t-k:t-1}$ at timestep $t$, we incorporate it into the dynamics prediction model $\hat{f}$ to improve its generalization ability on different dynamics by optimizing the objective function following (Lee et al., 2020; Seo et al., 2020; Janner et al., 2019):

$$\mathcal{L}_{\theta,\phi}^{pred} = -\frac{1}{N}\sum_{i=1}^{N} \log \hat{f}(s_{t+1}^{i}|s_t^i, a_t^i, g(\tau_{t-k:t-1}^i; \phi); \theta), \tag{1}$$

where $k$ is the length of transition segments, $t$ is the current timestep and $N$ is the sample size. In practice, we sub-sample a mini-batch of data from the whole dataset to estimate (1) and use stochastic gradient descent to update the model parameters.

However, as analyzed in Section 3.2, the vanilla prediction error (1) is not sufficient to capture environment-specified $Z$ of each environment, and even introduce redundant information into it. In order to eliminate the redundant information within transition segments and preserve the trajectory invariant information, we introduce a relational head (Patacchiola & Storkey, 2020) as a learnable function $h$ to pull factors $\hat{Z}$ from the same trajectory together and push away those from different trajectories. Concretely, the estimated $\hat{z}_{t-k:t-1}^i$ in a mini-batch will be firstly aggregated as pairs, $e.g.$ concatenate two factors as $[\hat{z}^i, \hat{z}^j]$, and the pairs having two factors from the same trajectory are seen as positives, and vice versa. Then the relational head $h$ parameterized by $\varphi$ takes a pair of aggregated factors as input to quantify the similarity of given two factors and returns a similarity score $\hat{y}$. To increase the similarity score $\hat{y}$ of positive pairs and decrease those negatives, we minimize the following objective:

$$\mathcal{L}_{\varphi,\phi}^{relation} = -\frac{1}{N(N-1)}\sum_{i=1}^{N}\sum_{j=1}^{N}\Big[y^{i,j}\cdot\log h([\hat{z}^i, \hat{z}^j]; \varphi)+(1-y^{i,j})\cdot\log\big(1-h([\hat{z}^i, \hat{z}^j]; \varphi)\big)\Big], \tag{2}$$

where $y^{i,j} = 1$ stands for positive pairs, and $y^{i,j} = 0$ stands for negatives. Because the positive pairs have two factors belonging to the same trajectory, optimizing (2) can increase the similarity of $\hat{Z}$s estimated from the same trajectory, and push away those factors estimated from different trajectories in their semantical space. Therefore, by optimizing (2), the information that is invariant within a trajectory will be encoded into $\hat{Z}$ and the redundant information in transition segments will be reduced. (2) can also be interpreted from the perspective of mutual information, if we regard the $\hat{Z}$s from the same trajectory as the positive pairs, optimizing (2) can be seen as maximizing the mutual information between $\hat{Z}$s from the same trajectory (Please refer to (Tsai et al., 2020) and Appendix A.5), and thus preserve the invariant information with the same trajectory. However, estimating trajectory invariant information is insufficient because the estimated $\hat{Z}$s in the same environment will also be pushed away, which may undermine the cluster compactness for estimated $\hat{Z}$s.

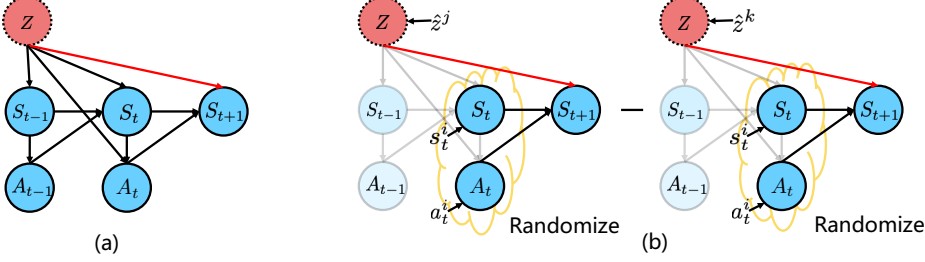

Figure 3: (a) The illustration of causal graph, and the red line denotes the direct causal effect from $Z$ to $S_{t+1}$. (b) The illustration of estimating the controlled causal effect.

## 3.4 INTERVENTIONAL PREDICTION

Because the environment id of a trajectory is unknown, we cannot directly optimize relational loss (2) to cluster $\hat{Z}$ within an environment. We propose an interventional prediction method to find the trajectories belonging to the same environment. Here we formalize the dynamics prediction model using a graphical causal model, and the causal graph is illustrated as Figure 3 (a), where the next state $S_{t+1}$ is caused by the current state $S_t$, action $A_t$ and $\hat{Z}$, and the dynamics prediction model $f$ represents the causal mechanism between them. Because $Z$ from the same environment should have

the same influence on the states, and thus they should have the same causal effect on the next state $S_{t+1}$ if given $S_t$ and $A_t$ under the causal framework. As such, we can find estimated $\hat{Z}$ belonging to the same environment by measuring the similarity of their causal effect on $S_{t+1}$. As Figure 3 (a) shows, there are multiple paths from $Z$ to $S_{t+1}$ and we roughly categorize them into two categories. The first category is the direct path between $Z$ and $S_{t+1}$ ( shown as read in Figure 3). The second category contains all the indirect paths where $Z$ influences $S_{t+1}$ via previous states and actions. However, because the mediator in other paths *e.g.* $S_t$, $A_t$, may amplify or reduce the causal effect of $Z$, we only consider the direct path from $Z$ to the next state(denote by the red line at Figure 3 (a)), which means that we need to block all paths with meditors from $\hat{Z}$ to $S_{t+1}$. By means of do-calculus (Pearl, 2000), we can estimate the direct causal effect of changing $Z = \hat{z}^j$ to $Z = \hat{z}^k$ on $S_{t+1}$ through calculating the controlled direct effect (CDE) (Pearl, 2013) by intervening mediators and $\hat{Z}$ :

$$CDE_{\hat{z}^j,\hat{z}^k}(s_t,a_t) = \mathbb{E}[S_{t+1}|do(S_t = s_t, A_t = a_t), do(Z = \hat{z}^j)] - \mathbb{E}[S_{t+1}|do(S_t = s_t, A_t = a_t), do(Z = \hat{z}^k)] \tag{3}$$

$$= \mathbb{E}[S_{t+1}|S_t = s_t, A_t = a_t, Z = \hat{z}^j] - \mathbb{E}[S_{t+1}|S_t = s_t, A_t = a_t, Z = \hat{z}^k], \tag{4}$$

where $do$ is the do-calculus (Pearl, 2000). There is no arrow entering $\hat{Z}$, so the do operator on $\hat{Z}$ can be removed. Also, since there is no confounder between the mediators $(S_t, A_t)$ and $S_{t+1}$, so we can remove the do operator of them as well, and the equation become as (4). Because the direct causal effects may differ for different values of $S_t$ and $A_t$, we should sample $S_t$ and $A_t$ independently of $Z$, *i.e.* sampling $S_t$ and $A_t$ (Pearl et al., 2016) uniformly to get the average controlled direct causal effect from $\hat{Z}$ to $S_{t+1}$. However, if we use the uniformly generated $S_t$ and $A_t$, the sampled distribution may differ from the training distribution, resulting in inaccurate the next state prediction. As such, we directly sample $S_t$ and $A_t$ from the observational data. For the convenience of optimization, we only use a mini-batch of $S_t$ and $A_t$ pairs $(s_t^i, a_t^i)$, and concatenate them with $\hat{z}^j$ and $\hat{z}^k$ to calculate the average controlled direct effect under $\hat{f}$:

$$ACDE_{\hat{z}^j,\hat{z}^k} = \frac{1}{N}\sum_{i=1}^{N}|CDE_{\hat{z}^j,\hat{z}^k}(s_t^i,a_t^i)|, \tag{5}$$

where $N$ is the batch size, $j$ and $k$ are the id of $\hat{Z}$ estimated from two transition segments. Specifically, because the factors $\hat{z}$ estimated from the same trajectory should be the same, and thus we minimize their controlled direct effect 5 as $\mathcal{L}^{dist}$ between them in the optimization process. Now we can use the calculated $ACDE_{\hat{z}^j,\hat{z}^k}$ as the semantic distance $d^{j,k}$ between estimated $\hat{z}^i$ and $\hat{z}^j$, and thus we can aggregate factors $\hat{Z}$ estimated from similar trajectories by the proposed relational head $h$. As such, we apply a transformation to convert distance metric $d^{j,k}$ to a similarity metric $w \in (0,1]$, which is $w^{j,k} = exp(\frac{-d^{j,k}}{\beta})$, where $\beta$ is a factor controlling the sensitivity of the distance metric. Specifically, because the scale and size of state varies in different tasks, *e.g.* 3 dims in Pendulum but 20 dims in Half-Cheetah, the optimal $\beta$ may vary in different task. As such, we apply the normalization in the distance metric $d$, *i.e.*, normalize $d$ with batch variance, to convert it as a relative distance within a single task, thus making the optimal $\beta$ stable in different tasks. Then we can directly aggregate similar trajectories by extending the loss function (2) as follows:

$$\mathcal{L}_{\varphi,\phi}^{i-relation} = -\frac{1}{N(N-1)}\sum_{i=1}^{N}\sum_{j=1}^{N}\Big[ [y^{i,j} + (1 - y^{i,j}) \cdot w^{i,j}] \cdot \log\ h([\hat{z}^i, \hat{z}^j]; \varphi)$$
$$+ (1 - y^{i,j}) \cdot (1 - w^{i,j}) \cdot \log\ (1 - h([\hat{z}^i, \hat{z}^j]; \varphi))\Big], \tag{6}$$

where the first term indicates that the factors from the different trajectories can be aggregated with the similarity weight $w$ and 1 those from the same trajectory, and the second term means that factors from different trajectories should be pushed with each other with weight $1 - w$. Similar to the analysis in section 3.3, optimizing the loss function (6) can increase the similarity between $\hat{Z}$ with weight $w$, and push away from them with the weight $1 - w$. Because $\hat{Z}$s estimated from the same environment have similar effects, these factors will be assigned with high similarities (estimated by the intervention operation of our paper). By simultaneously updating the prediction loss (1) and intervention relation loss 6, estimated $\hat{Z}$s within the same environment will be aggregated, and the learned dynamics function $\hat{f}$ can learn the modals of transition dynamics according to the $\hat{Z}$ in different clusters. The training procedure of our approach can refer to Algorithm process in Appendix A.2.

## 4 EXPERIMENTS

In this section, we conduct experiment to evaluate the performance of our approach by answering the following questions:

- Can our approach reduce the dynamics prediction errors in model-based RL? (Section 4.2.1)
- Can our approach promote the performance of model-based RL on environments with unseen dynamics? (Section 4.2.2)
- Can our approach learn the semantic meaningful dynamics change? (Figuer 1 and AppendixA.13)
- Is the similarity of $w$ measured by the intervention module reasonable? (Appendix A.7)
- Can solely relational learning improve the performance of model-based RL? (Section 4.3)

### 4.1 ENVIROMENTAL SETUP

**Implementation Details**   Our approach includes three learnable functions, including relational encoder, relational head and prediction head. All three functions are constructed with MLP and optimized by Adam (Kingma & Ba, 2014) with the learning rate 1e-3. During the training procedure, the trajectory segments are randomly sampled from the same trajectory to break the temporal correlations of the training data, which was also adopted by (Seo et al., 2020; Wang et al., 2020b; 2019). Specifically, the length of the transition segments, *i.e.*, $k$, is 10. All implementation details can be found in Appendix A.1.

**Datasets**   Following the previous methods (Lee et al., 2020; Seo et al., 2020), we perform experiments on a classic control task (Pendulum) from OpenAI gym (Brockman et al., 2016) and simulated robotics control tasks (HalfCheetah, Cripple-HalfCheetah, Ant, Hopper, Slim-Humanoid) from Mujoco physics engine (Todorov et al., 2012).

**Dynamics Settings**   To change the dynamics of each environment, we follow previous methods (Zhou et al., 2019; Packer et al., 2019; Lee et al., 2020; Seo et al., 2020) to change the environmental parameters (*e.g.* length and mass of Pendulum) and predefine them in the training and test environmental parameters lists. At the training time, we randomly sample the parameters from the training parameter list to train our relational context encoder and dynamics prediction model. Then we test our model on the environments with unseen dynamics sampled from the test parameter list. Specifically, the predefined parameters in the test parameter list are outside the training range. The predefined training and test parameter lists for each task are the same with (Lee et al., 2020), and all details are given in Appendix A.1.

**Planning**   Following (Lee et al., 2020; Seo et al., 2020), we use the model predictive model (MPC) (Maciejowski, 2002) to select actions based on learned dynamics prediction model, and assume that the reward functions of environments are known. Also, the cross-entropy method (CEM) (De Boer et al., 2005) is used to optimize action sequences for finding the best performing action sequences.

**Baselines**   We compare our approach with following state-of-the-art model-based RL methods on dynamics generalization. Also, to show the performance gap between our method and supervised dynamics generalization, we perform the method using true environment label to cluster $Z$.

- Probabilistic ensemble dynamics model (PETS) (Kurutach et al., 2018): PETS employs an probabilistic dynamics models to capture the uncertainty in modeling and planning.
- Meta learning based model-based RL (ReBAL and GrBAL) (Nagabandi et al., 2018b;a): These methods train a dynamics model by optimizing a meta-objective (Finn et al., 2017), and update the model parameters by updating a hidden with a recurrent model or by updating gradient updates at the test time.
- Context-aware dynamics model (CaDM) (Lee et al., 2020): This method design several auxiliary loss including backward and future states prediction to learn the context from transition segments.
- Trajectory-wise Multiple Choice Learning (TMCL) (Seo et al., 2020): This method is the state-of-the-art model-based RL method on dynamics generalization, which introduces the multi-choice learning to cluster environments. TMCL needs the adaptation in the test procedure, while our method does not, so we also report the performance of TMCL without adaptation in Figure 7 for the fair comparison.

- True Label: The method uses our relational head to cluster $\hat{Z}$ with the true environment label (not the ground-truth of $Z$). All hyperparameters are same with our method for the fair comparison.

## 4.2 PERFORMANCE COMPARISONS WITH BASELINES

### 4.2.1 PREDICTION ERROR COMPARISONS

We first evaluate whether the dynamics model trained by our methods can predict next-states more accurately or not. Figure 4 shows that the average dynamics prediction error of dynamics prediction models trained by three methods (CaDM (Lee et al., 2020), TMCL (Seo et al., 2020) and ours). We can see that the dynamics model trained by our relational intervention method has superior prediction performance over other state-of-the-art methods, achieving the lowest prediction errors on almost all six tasks. Specifically, the prediction errors of our model are lower than others by a large margin in Hopper and Pendulum, outperforming the state-of-the-art methods by approximately 10%.

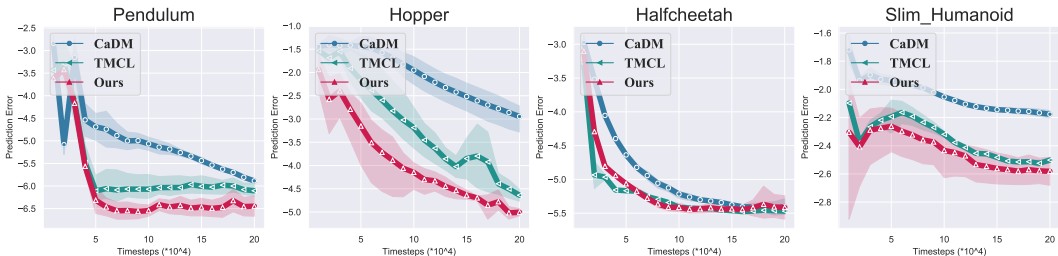

Figure 4: The average prediction errors of dynamics models on training environments during training process (over three times). Specifically, the x axis is the training timesteps and y axis is the $log$ value of average prediction prediction errors. More figures are given at Appendix A.8.

Table 1: The average rewards of baseline model-based RL methods and ours on test environments with unseen dynamics. Here we report the average rewards over three runs (ours is ten). Specifically, the results of methods with $*$ are from the paper (Lee et al., 2020).

|  | PETS* | ReBAL* | GrBAL* | CaDM | TMCL | Ours | ↑ Ratio |
|---|---|---|---|---|---|---|---|
| Pendulum | -1103 | -943.6 | -1137.9 | -713.95±21.1 | -691.2±93.4 | **-587.5**±64.4 | 15.0% |
| Ant | 965.883.5 | 63.0 | 44.7 | 1660±57.8 | 2994.9±243.8 | **3297.9**±159.7 | 10.1% |
| Hopper | 821.2 | 846.2 | 621 | 845.2±20.41 | 999.35±22.8 | **1057.4**±37.2 | 5.8% |
| HalfCheetah | 1720.9 | 52 | -69.1 | 5876.6±799.0 | 9039.6±1065 | **10859.2**±465.1 | 20.1% |
| C_HalfCheetah | 1572 | 868.7 | 2814 | 3656.4±856.2 | 3998.8±856.2 | **4819.3**±409.3 | 20.5% |
| Slim_Humanoid | 784.5 | 97.25 | -480.7 | 859.1±24.01 | 2098.7±109.1 | **2432.6**±465.1 | 15.9% |

### 4.2.2 PERFORMANCE COMPARISONS

Then we evaluate the generalization of model-based RL agents trained by our methods and baselines on test environments with unseen dynamics. Following the setting of (Seo et al., 2020), we perform experiments three runs (ours with 10 runs to reduce random errors), and give the mean of rewards at Table 1. We can see that the meta-learning based methods (Nagabandi et al., 2018b;a) do not perform better than vanilla PETS (Kurutach et al., 2018), while methods (Lee et al., 2020; Seo et al., 2020) that aim to learn a generalized dynamics prediction model are superior to others significantly. Among which our approach achieves the highest rewards on all six tasks among all methods. Figure 5 shows the mean and standard deviation of average rewards during the training procedure, indicating that the performance of our methods is better than the other two methods consistently at the training time, which is sufficient to show the superiority of our method over other methods. A fair comparison between TMCL (no adaptation) and our method can be found at Appendix A.6. In addition, we observe that our method achieves comparable results with the method directly cluster $\hat{Z}$ using the truth environment label, which indicates that our intervention module actually can assign high similarities into $\hat{Z}$s estimated from the same environment in an unsupervised manner. We also observe the same results in the similarity visualization in the Appendix A.7, where we find that $\hat{Z}$s from the same environment are assigned significant higher similarities than those pairs from different environments.

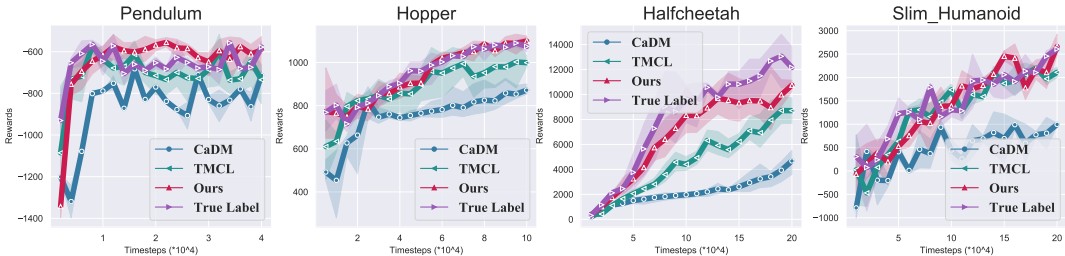

Figure 5: The average rewards of trained model-based RL agents on unseen test environments. The results show the mean and standard deviation of returns averaged over three runs. The fair comparison between TMCL (no adaptation) and our method can be found in Appendix A.6

### 4.3 ABLATION STUDY

In this section, we evaluate the effect of the proposed relation head and intervention prediction on the generalization improvement, respectively. Because the intervention prediction is based on the relational head, we compare the performance of our approach with and without the intervention. As Figure 6a and 6b show, after incorporating the relational head and intervention prediction, the performance of model-based agents and the generalization of the dynamics prediction model are both improved. However, although the model without the intervention module has lower prediction errors in the Pendulum task, it also has lower rewards than the whole model. One possible reason is that the Pendulum is simple for the dynamics prediction model to learn, and thus the dynamics prediction model with the vanilla relational head is a little over-fitting on the training environments (Please refer to prediction errors on test environments are given in Appendix A.9), limiting the performance improvement. This phenomenon confirms the importance of our intervention prediction on reducing the trajectory-specified redundant information.

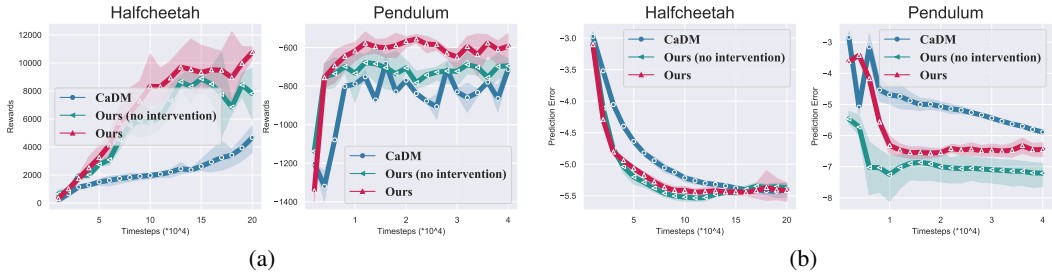

(a)                                        (b)

Figure 6: (a) The average rewards of trained model-based RL agents on unseen environments. The results show the mean and standard deviation of returns averaged over three runs. (b) The average prediction errors over the training procedure. Prediction errors on test environments are given in Appendix A.9

## 5 CONCLUSION

In this paper, we propose a relational intervention approach to learn a generalized dynamics prediction model for dynamics generalization in model-based reinforcement learning. Our approach models the dynamics change as the variation of environment-specified factor $\mathcal{Z}$ and explicitly estimates $\mathcal{Z}$ from past transition segments. Because environment label is not available, it is challenging to extract $\mathcal{Z}$ from transition segments without introducing additional redundant information. We propose an intervention module to identify the probability of two estimated factors belonging to the same environment, and a relational head to cluster those estimated $\hat{Z}$s are from the same environments with high probability, thus reducing the redundant information unrelated to the environment. By incorporating the estimated $\hat{Z}$ into the dynamics prediction process, the dynamics prediction model has a stronger generalization ability against the change of dynamics. The experiments demonstrate that our approach can significantly reduce the dynamics prediction error and improve the performance of model-based agents on new environments with unseen dynamics.

## 6    ACKNOWLEDGE

Mr Jiaxian Guo is supported in part by Australian Research Council Projects FL-170100117 and LE-200100049. Dr Mingming Gong is supported by Australian Research Council Project DE210101624.

## 7    ETHICS STATEMENT

Our paper provides a method to generalize the agent trained by the model-based reinforcement learning into new environments with unseen transition dynamics, which may significantly improve the robustness of trained agents in the complex real-world environment, thus broadening the application scenarios of robots trained by reinforcement learning. Although it is far away to apply such an algorithm to real-world applications, we still need to prevent the algorithm from being applied in military areas.

## 8    REPRODUCIBILITY STATEMENT

We have run our experiments over three runs  (ours with 10 runs to reduce random errors) to reduce random errors, and public hyperparameters used in our experiments. The codes of this method are available at `https://github.com/CR-Gjx/RIA`.

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

# A  APPENDIX

**We promise that we will public all codes after the acceptance of this paper.**

## A.1  ENVIRONMENTAL SETTINGS

We follow the environmental settings of Lee et al. (2020) in dynamics generalization. The details of settings are given as follows:

- **Pendulum** We modify the mass $m$ and the length $l$ of Pendulum to change its dynamics.

- **Half-Cheetah** We modify the mass of regid link $m$ and the damping of joint $d$ of Half-Cheetah agent to change its dynamics.

- **Crppled_Cheetah** We cripple the id of leg $c$ of Half-Cheetah agent to change its dynamics.

- **Ant** We modify the mass of ant's leg $m$ to change its dynamics. Specifically, we modify two legs by multiplying its original mass with $m$, and others two with $\frac{1}{m}$.

- **Slim_Humanoid** We modify the mass of rigid link $m$ and the dampling of joint $d$ of the Slim_Humanoid agent to change its dynamics.

- **Hopper** We modify the mass of $m$ of the Hopper agent to change its dynamics.

The training and test modified parameter list can be found at the Table 2.

Table 2: The environmental settings in our paper.

|  | Training Parameter List | Test Parameter List | Episode Length |
|---|---|---|---|
| Pendulum | $m \in \{0.75,0.8,0.85,0.90,0.95,$ $1,1.05,1.1,1.15,1.2,1.25\}$ $l \in \{0.75,0.8,0.85,0.90,0.95,$ $1,1.05,1.1,1.15,1.2,1.25\}$ | $m \in \{0.2,0.4,0.5,0.7,$ $1.3,1.5,1.6,1.8\}$ $l \in \{0.2,0.4,0.5,0.7,$ $1.3,1.5,1.6,1.8\}$ | 200 |
| Half-Cheetah | $m \in \{0.75,0.85,1.00,1.15,1.25\}$ $d \in \{0.75,0.85, 1.00,1.15,1.25\}$ | $m \in \{0.2,0.3,0.4,0.5,$ $1.5,1.6,1.7,1.8\}$ $d \in \{0.2,0.3,0.4,0.5,$ $1.5,1.6,1.7,1.8\}$ | 1000 |
| C_Cheetah | $c \in \{0,1,2,3\}$ | $c \in \{4,5\}$ | 1000 |
| Ant | $m \in \{0.85,0.90,0.951.00\}$ | $m \in \{0.20,0.25,0.30,0.35,0.40,$ $0.45,0.50,0.55,0.60\}$ | 1000 |
| Slim_Humanoid | $m \in \{0.80,0.90,1.00,1.15,1.25\}$ $d \in \{0.80,0.90,1.00,1.15,1.25\}$ | $m \in \{0.40,0.50,0.60,0.70,$ $1.50,1.60,1.70,1.80\}$ $d \in \{0.40,0.50,0.60,0.70,$ $1.50,1.60,1.70,1.80\}$ | 1000 |
| Hopper | $m \in \{0.5, 0.75, 1.0, 1.25, 1.5\}$ | $m \in \{0.25, 0.375, 1.75, 2.0\}$ | 500 |

## A.2  ALGORITHM

The training procedure is give at Algorithm 1.

## A.3  TRAINING DETAILS

Similar to the Lee et al. (2020), we train our model-based RL agents and relational context encoder for 20 epochs, and we collect 10 trajectories by a MPC controller with 30 horizon from environments at each epoch. In addition, the cross entropy method (CEM) with 200 candidate actions is chosen as the planing method. Specifically, the batch size for each experiment is 128, $\beta$ is 6e-1. All module are learned by a Adam optimizer with 0.001 learning rate.

---

**Algorithm 1** The training algorithm process of our relational intervention approach

---

Initialize parameters of relational encoder $\phi$, dynamics prediction model $\theta$ and relational head $\varphi$
Initialize dataset $\mathcal{B} \leftarrow \emptyset$
**for** Each Iteration **do**
    sample environments $\mathcal{M}^i$ from training environments $\{\mathcal{M}_i^{tr}\}_{i=0}^K$         $\triangleright$ Collecting Data
    **for** $T = 1$ to TaskHorizon **do**
        Get the estimation of the environment-specified factor $\hat{z}_{t-k:t-1}^i = g(\tau_{t-k:t-1}^i; \phi)$
        Collect $(s_t, a_t, s_{t+1}, r_t, \tau_{t-k:t-1}^i)$ from $\mathcal{M}^i$ with dynamics prediction model $\theta$
        Update $\mathcal{B} \leftarrow \mathcal{B} \cup (s_t, a_t, s_{t+1}, r_t, \tau_{t-k:t-1}^i)$
    **end for**
    **for** Each Dynamics Training Iteration **do**         $\triangleright$ Update $\phi$,$\theta$ and $\varphi$
        **for** $k = 1$ to K **do**
            Sample data $\tau_{t-k:t-1}^{i,b,P}$ , $\tau_{t:M}^{i,b,K}$ and $\tau_{t-k:t-1}^{j,b,P}$ , $\tau_{t:M}^{j,b,K}$ with batch size B,from $\mathcal{B}$
            Get the estimation of the environment-specified factor $\hat{z}_{t-k:t-1}^{i,B,,P} = g(\tau_{t-k:t-1}^{i,B,,P}; \phi)$ and
            $\hat{z}_{t-k:t-1}^{ij,B,,P} = g(\tau_{t-k:t-1}^{j,B,,P}; \phi)$
            Estimate the probability $w$ of $\hat{z}_{t-k:t-1}^{i,B,,P}$ and $\hat{z}_{t-k:t-1}^{j,B,,P}$ belonging to the same environment.
            Compute the total loss
            $\mathcal{L}^{tot} = \mathcal{L}_{\phi,\theta}^{pred}(\tau_{t:M}^{i,B,,K}, \hat{z}_{t-k:t-1}^{i,B,,P}) + \mathcal{L}_{\phi,\varphi}^{i-relation}(\hat{z}_{t-k:t-1}^{i,B,,P}) + \mathcal{L}_{\phi,\theta}^{dist}(\tau_{t:M}^{i,B,K}, \hat{z}_{t-k:t-1}^{i,B,P})$
            Update $\theta$ , $\phi$ , $\varphi \leftarrow \nabla_{\theta,\phi\,\varphi} \frac{1}{B} \mathcal{L}^{tot}$
        **end for**
    **end for**
**end for**

---

## A.4 NETWORK DETAILS

Similar to the Lee et al. (2020), the relational encoder is constructed by a simple 3 hidden-layer MLP, and the output dim of environmental-specific vector $\hat{z}$ is 10. The relational head is modelled as a single FC layer. The dynamics prediction model is a 4 hidden-layer FC with 200 units.

## A.5 CONNECTION BETWEEN RELATION LOSS AND MUTUAL INFORMATION

Given a pair of data $(x, y) \in \mathcal{X} \times \mathcal{Y}$, we donote the joint distribution of $X$ and $Y$ are $P_{XY}$, and their marginal distributions are $P_X$ and $P_Y$, respectively. By definition, the mutual information between $X$ and $Y$ is:

$$I(X;Y) = \mathbb{E}_{P_{XY}}[\log(\frac{p(x,y)}{p(x)p(y)})] \tag{7}$$

To estimate mutual information between $X$ and $Y$, (Tsai et al., 2020) proposes a probabilistic classifier method. Concretely, we can use a Bernoulli random variable $C$ to classify one given data pair $(x, y)$ from the joint distribution $P_{XY}$ ($C = 1$) or from the product of marginal distribution $P(X)P(Y)$ ($C = 0$) . Therefore, the mutual information $I(X;Y)$ between $X$ and $Y$ can be rewrite as:

$$
\begin{aligned}
I(X;Y) &= \mathbb{E}_{P_{XY}}[\log(\frac{p(x,y)}{p(x)p(y)})] \\
&= \mathbb{E}_{P_{XY}}[\log(\frac{p(x,y|C=1)}{p(x,y|C=0)})] \\
&= \mathbb{E}_{P_{XY}}[\log(\frac{p(C=0)P(C=1|x,y)}{p(C=1)P(C=0|x,y)})]
\end{aligned}
\tag{8}
$$

Obviously, $\frac{p(C=0)}{p(C=1)}$ can be approximated by the sample size, *i.e.* $\frac{n_{P_X P_Y}}{n_{P_{XY}}}$, while $\frac{P(C=1|x,y)}{P(C=0|x,y)}$ can be measured by a classifier $h(C|x,y)$, and it can be learned by our relation loss with relational head $h$:

$$\mathcal{L}_{\varphi,\phi}^{relation} = -\Big[ C \cdot \log\, h([x,y]; \varphi) + (1-C) \cdot \log\,(1 - h([x,y]; \varphi)) \Big], \tag{9}$$

where $C = 1$ if the given pair $(x, y)$ is from the joint distribution $P_{XY}$, and $C = 0$ if the given pair $(x, y)$ is from the product of the marginal distributions $P_X P_Y$. Because $\frac{p(C=0)}{p(C=1)}$ tend to be a constant,

optimizing our relation loss is actually estimating the mutual information $I(X;Y)$ between $X$ and $Y$. As such, if we regard the pairs of $(\hat{z})$ from the same trajectory/environment as positive pairs, and others are negative pairs, optimizing 2 is actually maximizing the mutual information between $(\hat{z})$ from the same trajectory/environment, and thus preserve the trajectory/environment invariant information. If the readers are interested in the concrete bound about this method to estimate mutual information, please refer to (Tsai et al., 2020).

## A.6 FAIR COMPARISON WITH TMCL

Because TMCL needs an adaptation process when deploying it into the real world while our method does not. For the fair comparison and show the significance of our method over TMCL, we test the performance of TMCL with no adaptation, and show the results below:

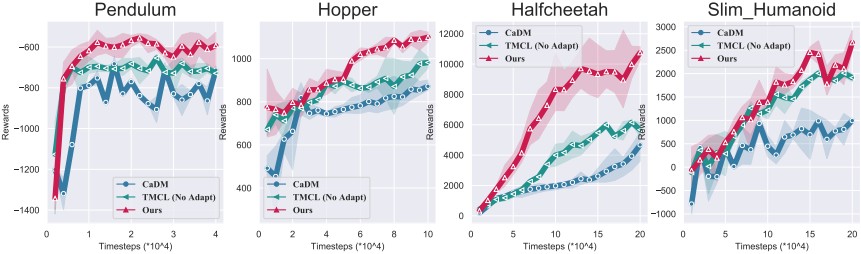

Figure 7: The estimated similarities between $\hat{Z}$s (learned by the model without Intervention module) from different environments and anchors (mass=1) on different tasks, where the box represent the range of 50% samples, the line in the middle of a box denotes the average similarity and the top/bottom lines denote the max/min similarities.

We can see that the average returns of TMCL without adaptation are significantly lower than ours, especially for the classic control task Halfcheetah. The experimental comparison with TMCL without adaptation is direct evidence to support our claim: environment-separated $Z$s is important for the generalization of dynamics functions, and our method can significantly outperform baselines in zero-shot unseen test environments with different dynamics. Specifically, the performance of TMCL with no adaptation is still superior to the CaDM, this is because TMCL uses the invariance of Z within a trajectory (Z should predict other states within a trajectory in TMCL), which is similar to our paper with no intervention module.

## A.7 SIMILARITIES VISUALIAZTION

To evaluate the correctness of the estimated similarity of our intervention module, we use a $\hat{z}^i$ estimated from the environment where the mass is 1 as the anchor, and randomly sample 200 $\hat{z}^j$ estimated from different environments (including mass = 1). Then we calculate the similarity between anchor $\hat{z}^i$ and $\hat{z}^j$, and visualize the similarities according to their environments. As Figure 8 shows, $\hat{z}^j$s belonging to the same environment with the anchor $\hat{z}^i$ have significant higher similarities than those belonging to other environments, and even higher than 0.8 in some tasks (all are higher than 0.6), which shows that our intervention module can successfully identify whether two $\hat{z}$s from the same environment or not.

To study the role of the intervention module, we also visualize the similarity of $\hat{z}^i$ learned by the model without the intervention module, and the results are given as Figure 9. Figure 9 shows that many contexts from different environments still have high similarities. This indicates that the existing relational learning cannot separate environment-specified factors Zs. By contrast, after incorporating the intervention module, the contexts from different environments have significantly smaller similarities than those from the same environments. The comparison between Figures 8 and 9 directly shows that our intervention module is valuable to predict whether two contexts are from the same environment or not.

## A.8 PREDICTION ERRORS ON TRAING ENVIRONMENTS

The prediction errors of each method on training environment are given at Figure 10.

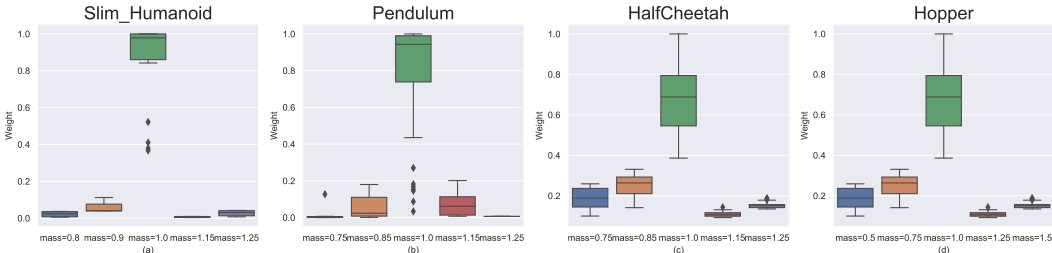

Figure 8: The estimated similarities between $\hat{Z}$s (learned by the model with Intervention module) from different environments and anchors (mass=1) on different tasks, where the box represent the range of 50% samples, the line in the middle of a box denotes the average similarity and the top/bottom lines denote the max/min similarities.

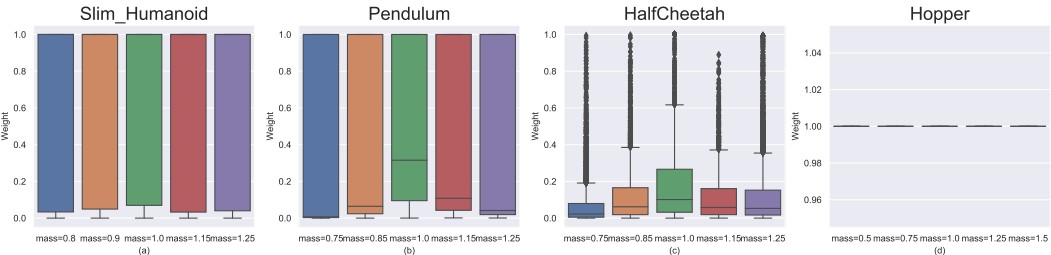

Figure 9: The estimated similarities between $\hat{Z}$s (learned by the model **without** Intervention module) from different environments and anchors (mass=1) on different tasks, where the box represent the range of 50% samples, the line in the middle of a box denotes the average similarity and the top/bottom lines denote the max/min similarities.

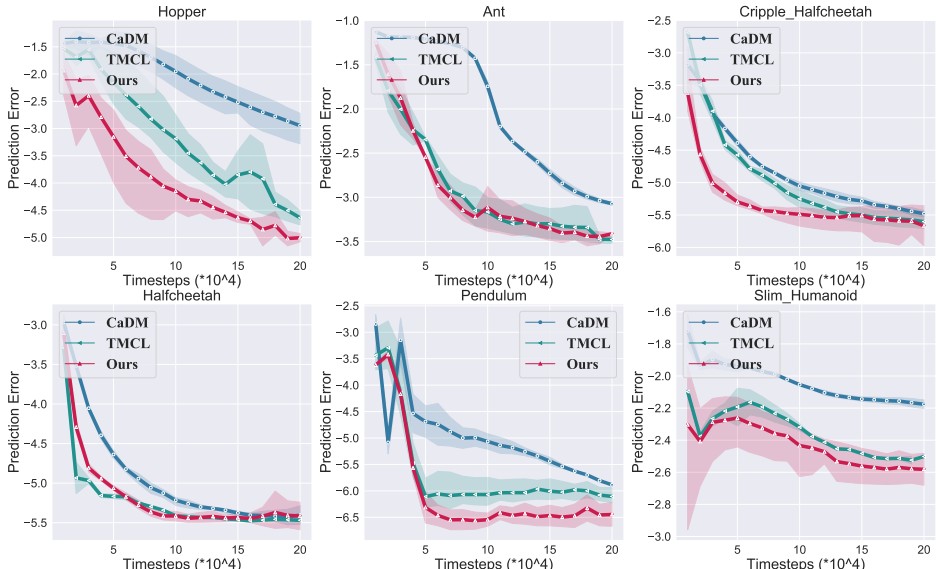

Figure 10: The average prediction errors of dynamics models on training environments during training process (over three times). Specifically, the x axis is the training timesteps and y axis is the $log$ value of average prediction prediction errors. More figures are given at Appendix A.8.

## A.9 PREDICTION ERRORS ON TEST ENVIRONMENTS

The prediction errors of each method on test environments are given at Table 3. Specifically, we test each test environment 10 times, and plot the average prediction error to reduce random errors (Figure 11).

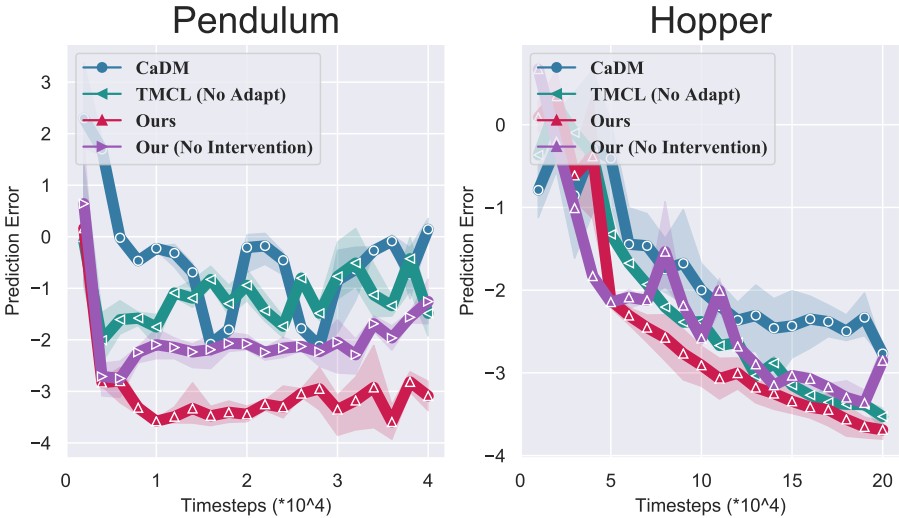

Figure 11: The average prediction errors of dynamics models on test environments during training process (over three times). Specifically, the x axis is the training timesteps and y axis is the average *log* value of prediction prediction errors.

Table 3: The prediction errors of methods on test environments

|  | CaDM (Lee et al., 2020) | TMCL (Seo et al., 2020) | Ours |
|---|---|---|---|
| Hopper | $0.0551 \pm 0.0236$ | $0.0316 \pm 0.0138$ | $\mathbf{0.0271 \pm 0.0011}$ |
| Ant | $0.3850 \pm 0.0256$ | $0.1560 \pm 0.0106$ | $\mathbf{0.1381 \pm 0.0047}$ |
| C_Halfcheetah | $0.0815 \pm 0.0029$ | $0.0751 \pm 0.0123$ | $\mathbf{0.0525 \pm 0.0061}$ |
| HalfCheetah | $0.6151 \pm 0.0251$ | $1.0136 \pm 0.6241$ | $\mathbf{0.4513 \pm 0.2147}$ |
| Pendulum | $0.0160 \pm 0.0036$ | $0.0130 \pm 0.0835$ | $\mathbf{0.0030 \pm 0.0012}$ |
| Slim_Humanoid | $0.8842 \pm 0.2388$ | $0.3243 \pm 0.0027$ | $\mathbf{0.3032 \pm 0.0046}$ |

## A.10 PREDICTION ERRORS ON SPECIFIED ENVIRONMENT

The prediction errors of each method on specified environment are given at Table 4, 5 and 6.

Table 4: The prediction errors of methods on specified environment of Hopper Task.

| mass | CaDM (Lee et al., 2020) | TMCL (Seo et al., 2020) | Ours |
|---|---|---|---|
| 0.25 | $0.0443 \pm 0.0049$ | $0.0294 \pm 0.0131$ | $\mathbf{0.0120 \pm 0.0025}$ |
| 1.75 | $0.0459 \pm 0.0006$ | $0.0131 \pm 0.0138$ | $\mathbf{0.0132 \pm 0.0013}$ |

Table 5: The prediction errors of methods on specified environment of Ant Task.

| mass | CaDM (Lee et al., 2020) | TMCL (Seo et al., 2020) | Ours |
|---|---|---|---|
| 0.30 | $0.0928 \pm 0.0019$ | $0.0910 \pm 0.0200$ | $\mathbf{0.0669 \pm 0.0040}$ |
| 0.50 | $0.1013 \pm 0.0057$ | $0.0887 \pm 0.0212$ | $\mathbf{0.0671 \pm 0.0034}$ |

Table 6: The prediction errors of methods on specified environment of Slim_Humanoid Task.

| mass | CaDM (Lee et al., 2020) | TMCL (Seo et al., 2020) | Ours |
|---|---|---|---|
| 0.50 | $0.1614 \pm 0.0165$ | $0.1860 \pm 0.0040$ | $\mathbf{0.1282 \pm 0.0295}$ |
| 0.70 | $0.1512 \pm 0.0152$ | $0.1550 \pm 0.0186$ | $\mathbf{0.1236 \pm 0.0162}$ |
| 1.50 | $0.1601 \pm 0.0202$ | $0.1873 \pm 0.0087$ | $\mathbf{0.1444 \pm 0.0233}$ |
| 1.70 | $0.1439 \pm 0.02029$ | $0.1688 \pm 0.01032$ | $\mathbf{0.1217 \pm 0.0206}$ |

## A.11 THE AVERAGE RETURNS ON TEST ENVIRONMENTS DURING TRAINING PROCESS

The average returns on test environments during training process are given at Figure 12.

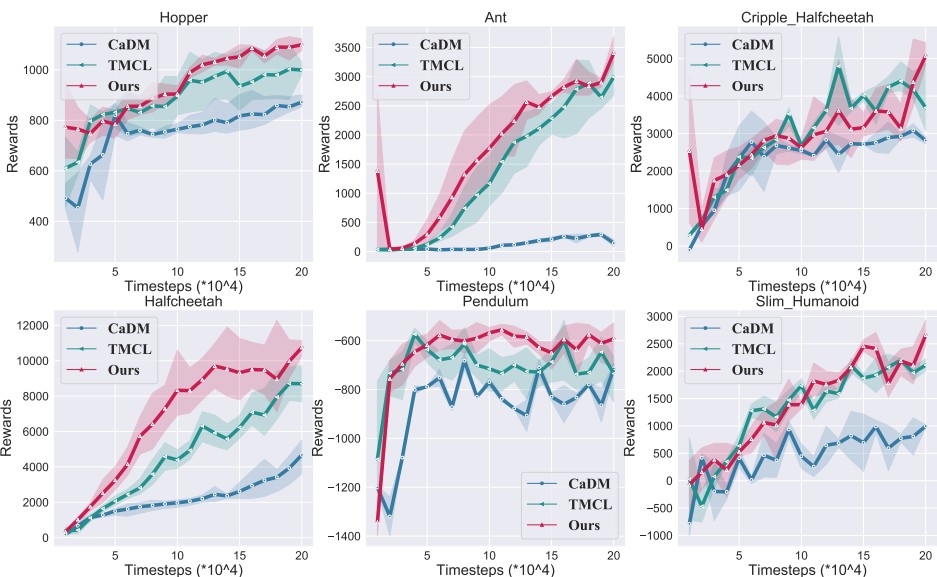

Figure 12: The average rewards of trained model-based RL agents on unseen environments. The results show the mean and standard deviation of returns averaged over three runs.

## A.12 QUANTITATIVE CLUSTERING PERFORMANCE COMPARISON

To quantitatively evaluate the $\hat{Z}$s' clustering performance, we use K-means algorithm to predict each $Z$'s environment id, and compare them with the true environment id. The details are provided in demo of K-means and evaluation metrics. The results are given at below. Specifically, TMCL has lower clustering performances than CaDM, but TMCL still has higher returns on test environments than CaDM. This is because TMCL clusters environments via multiplying dynamics functions rather than separating Zs.

Table 7: Quantitatively clustering evaluation results of $\hat{Z}$ on Pendulum.

|  | homo | compl | v-meas | ARI | AMI |
|---|---|---|---|---|---|
| CaDM | 1 | 0.655 | 0.627 | 0.516 | 0.599 |
| TMCL | 0 | 0.298 | 0.217 | 0.088 | 0.165 |
| Ours (no Intervention) | 0 | 0.768 | 0.762 | 0.760 | 0.653 |
| Ours | **1** | **0.932** | **0.932** | **0.937** | **0.931** |

Table 8: Quantitatively clustering evaluation results of $\hat{Z}$ on Halfcheetah.

|  | homo | compl | v-meas | ARI | AMI |
|---|---|---|---|---|---|
| CaDM | 0 | 0.262 | 0.260 | 0.203 | 0.257 |
| TMCL | 0 | 0.239 | 0.165 | 0.051 | 0.126 |
| Ours (no Intervention) | 0 | 0.368 | 0.362 | 0.265 | 0.353 |
| Ours | 0 | **0.416** | **0.411** | **0.312** | **0.405** |

According to the quantitative clustering performance measures, we can see that the clustering performance of our method is superior to baselines by a large margin, and the results are consistent with the performance on the test environments.

Table 9: Quantitative clustering evaluation results of $\hat{Z}$ on Slim_Humanoid.

|       | homo | compl | v-meas | ARI | AMI |
|-------|------|-------|--------|-----|-----|
| CaDM  | 0    | 0.046 | 0.045  | 0.027 | 0.042 |
| TMCL  | 0    | 0.002 | 0.002  | 0.000 | 0.000 |
| Ours  | 0    | **0.055** | **0.052** | **0.037** | **0.058** |

Table 10: Quantitative clustering evaluation results of $\hat{Z}$ on Cripple_Halfcheetah.

|       | homo | compl | v-meas | ARI | AMI |
|-------|------|-------|--------|-----|-----|
| CaDM  | 1    | 0.733 | 0.716  | 0.686 | 0.701 |
| TMCL  | 0    | 0.253 | 0.000  | 0.000 | 0.000 |
| Ours  | **1** | **0.853** | **0.851** | **0.860** | **0.849** |

Table 11: Quantitative clustering evaluation results of $\hat{Z}$ on Hopper.

|       | homo | compl | v-meas | ARI | AMI |
|-------|------|-------|--------|-----|-----|
| CaDM  | 0    | 0.019 | 0.018  | 0.010 | 0.015 |
| TMCL  | 0    | 0.023 | 0.008  | 0.000 | 0.003 |
| Ours  | 0    | **0.130** | **0.108** | **0.049** | **0.089** |

## A.13 VISUALIZATION

## A.14 T-SNE VISUALIZATION

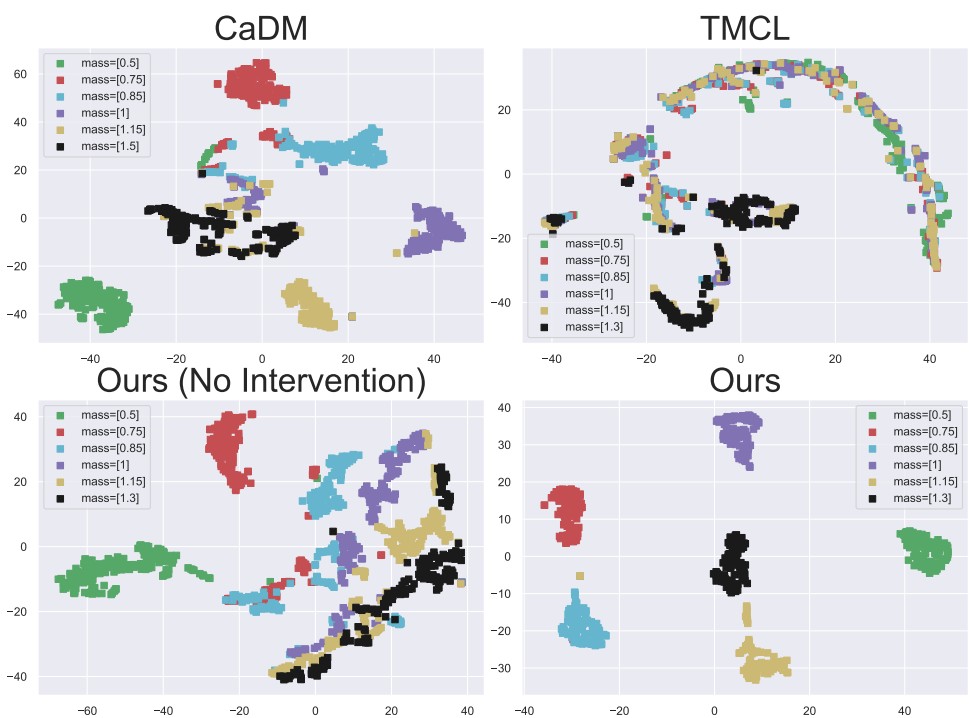

Figure 13: The T-SNE visualization of estimated context (environmental-specific) vectors in the **Pendulum** task, where mass = 0.5 and mass = 1.3 are from test environments.

## A.15 PCA VISUALIZATION

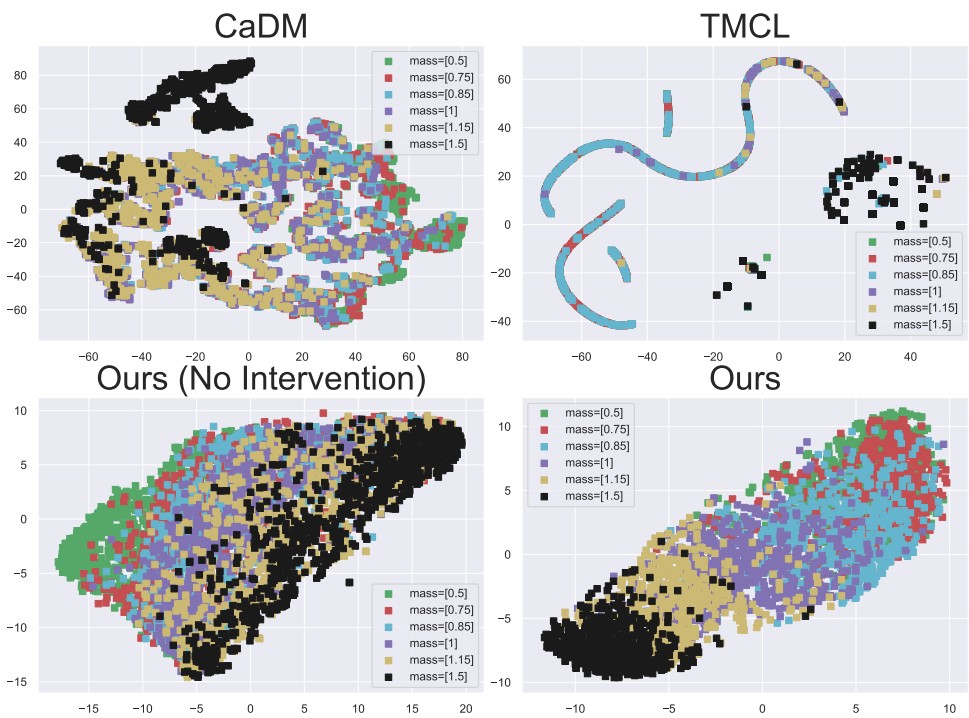

Figure 14: The T-SNE visualization of estimated context (environmental-specific) vectors in the **Halfcheetah** task, where mass = 0.5 and mass =1.5 are from test environments.

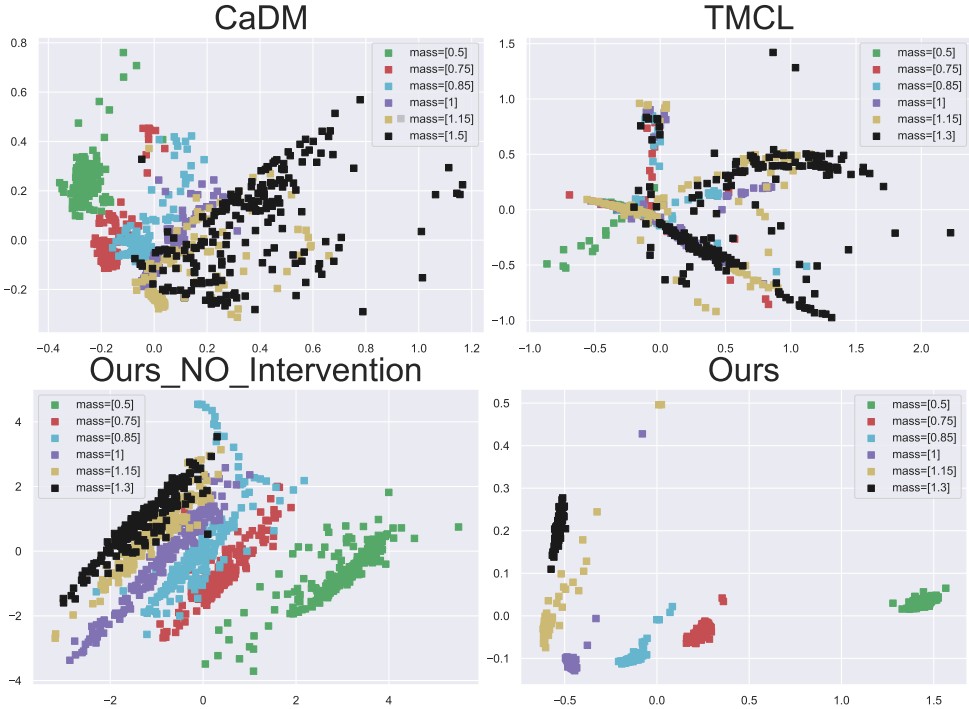

Figure 15: The PCA of estimated context (environmental-specific) vectors in **Pendulum** task, where mass = 0.5 and mass =1.3 are from test environments.

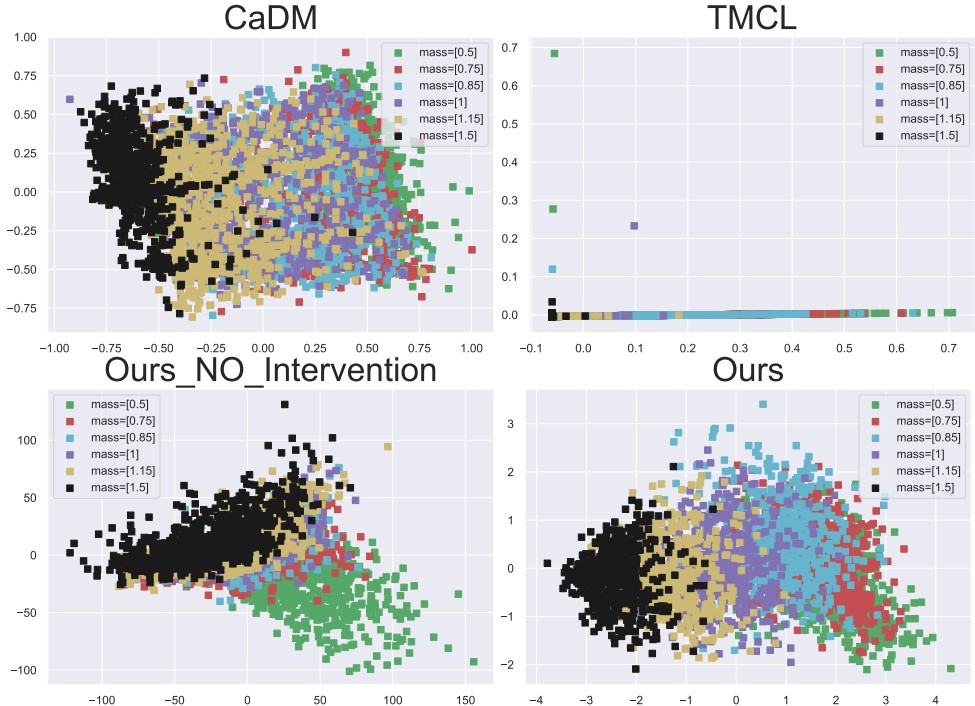

Figure 16: The PCA of estimated context (environmental-specific) vectors in **HalfCheetah** task, where mass = 0.5 and mass =1.5 are from test environments.

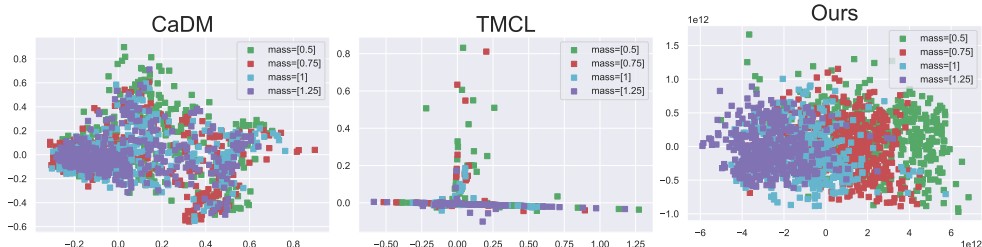

Figure 17: The PCA of estimated context (environmental-specific) vectors in the **Hopper** task.

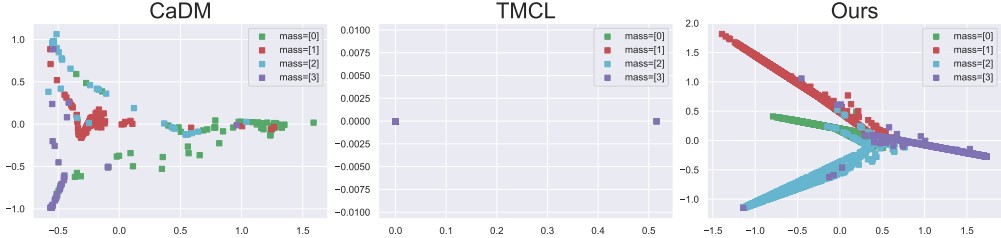

Figure 18: The PCA of estimated context (environmental-specific) vectors in the **Cripple_Halfcheetah** task.

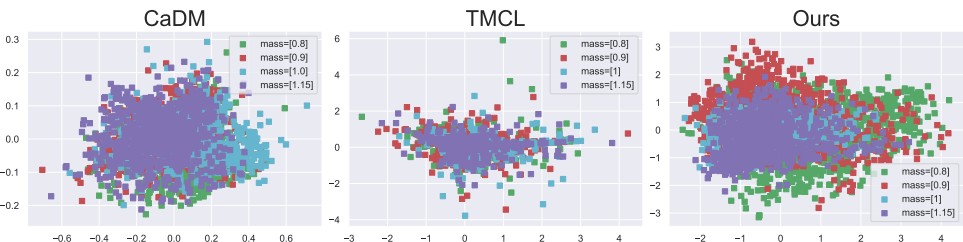

Figure 19: The PCA of estimated context (environmental-specific) vectors in the **Slim_Humanoid** task.

