# OpenReview forum: "A Relational Intervention Approach for Unsupervised Dynamics Generalization in Model-Based Reinforcement Learning"
_ICLR.cc/2022/Conference — ICLR 2022 Poster_

### Official Review · Reviewer_3e12 · 2021-10-28

**Correctness:** 4
**Technical Novelty And Significance:** 3
**Empirical Novelty And Significance:** 3
**Recommendation:** 8
**Confidence:** 4

**Main Review:**

# Strengths
- The problem statement identified by the paper is clear and compelling, which motivates the paper well.
- The method is interesting and novel, and seems well-designed to tackle the problem being addressed
- The empirical results and visualisations both support the claims that this method is performing better than previous SOTA methods due to the better clustering of contexts.

# Weaknesses
- The notation in the last paragraph of page 2 is confusing - what's the difference between Z and \hat Z?
- In general, I found the explanation of the method difficult to parse at first - it took me several passes to put all the pieces together, understand the justification for each piece and how it fulfilled it's role. I think that's partly due to slightly confusing notation (see above point) which isn't well defined, and also due to the complexity of the method. In some sense this can't be avoided, but it's still worth mentioning here.
- The biggest weakness of the paper (although I don't think it prevents acceptance at all) is that the method is a fairly incremental improvement to previous methods, and hence isn't highly novel. There's not much that can be done for this, unless additional benchmarks were created and assessed on (which is clearly a lot of work).

# Questions
- In the PCA visualisations, how are the different dots of the same colour generated? Is from different subsequences of transitions from a single episode, or multiple episodes, or both? It would be interesting to see how the context prediction evolves over an episode, and whether it converges over time as more information is gathered within the episode (as we might expect it to).
- The relational encoder take in a fixed number of past transitions, is that correct? How does this work for time-steps near the beginning of the episode, where not enough transitions are available? Have you investigated using a recurrent encoder which processes all the information in the episode?
- What's L_dist in the loss function in Algorithm 1?

# Suggestions for Improvement
- AugWM missing from related work: https://arxiv.org/abs/2104.05632, and seems relevant
- It would be interesting to visualise clustering of masses which are in testing distribution as well, to see how well the relational embedding generalises.
- The justification that their method improves through better clustering is only supported qualitatively through visualisation. It would be great to see a quantitative measure of how good the clustering is in the latent space for different methods, and then seeing if this correlates with evaluation performance or prediction error across the range of methods and environments (or even across hyperparameter options for each method).
- It would be good to see a plot of average prediction error on test environment during training, to see how that changes as the methods train.
- You provide some intuition as to why the ablation with no intervention has lower prediction error but lower performance in 4.3, but you could test this explicitly by plotting the test prediction error and seeing whether it's higher for the no-intervention ablation
- PCA is quite a limited visualisation technique for clustering - have you considered other techniques such as t-SNE or UMAP?

**Summary Of The Paper:**

This work tackles the problem of unsupervised dynamics generalisation in model-based reinforcement learning, improving approach introduced by additional of inferring a latent context which conditions the dynamics model and captures the variation in dynamics between environments. They introduce a set of auxiliary losses based on relational intervention and causal reasoning to encourage the inferred context to be the same in trajectories from the same environment (even when the environment identification is unknown) through inferring which environments are the same and weighting the context-similarity loss by this environment similarity. The method shows improved prediction error and test reward on a range of continuous control tasks compared to SOTA baselines from MBRL, Meta-RL and dynamics generalisation MBRL specifically. Visualisations qualitatively show that their method clusters transition segments from the same trajectory together better than previous methods, which could explain their improved performance.

**Summary Of The Review:**

I think the paper is well-motivated, presents and interesting and novel method and demonstrates that method's improved performance on the problem being targeted, and hence deserves acceptance. I think the visualisations and ablations are useful qualitative evidence for the benefit of the method, but additional visualisations, or more quantitative investigation of the clustering hypothesis would be beneficial. Also, the explanation of the method is sometimes difficult to understand. The method is only an incremental improvement on previous work, which is necessary to perform but limits it's impact and novelty.

EDIT: After seeing the revisions and additional experiments, I have raised my confidence from 3 to 4.

---

> ### Author Response · Authors · 2021-11-20
> **Response to Reviewer 3e12**
>
> Thanks for your constructive comments which are useful to improve our paper. Following your suggestion, we conduct more visualization experiments and reorganize our contributions in the revised paper. We hope that our response can resolve your concern. If you have further concerns, we are happy to follow up.
>
> **1. What's the difference between Z and $\hat Z$?**
>
> A: Thanks. $Z$ is the ground truth of the unobserved environment-specified factor, and $\hat{Z}$ is its estimation. Because $Z$ is unobservable, we learn $\hat{Z}$ from the historical transition segments and use it as a surrogate of $Z$. Following your suggestion, we have emphasized it in the revised paper.
>
>
>
> **2. In general, I found the explanation of the method difficult to parse, and the method is a fairly incremental improvement to previous methods, and hence isn't highly novel.**
>
> A: Thanks. Accordingly, we re-organize our contributions below:
>
> a)	We empirically demonstrate that both CaDM [2] and TMCL [1] fail to separate Zs from different environments (cf. Appendix A.4) when environment id is not available.
>
> b)	To eliminate the irrelevant information encoded in $\hat{Z}$, we (1) introduce controllable direct effect (CDE) to the RL community, (2) estimate the direct causal effect between $\hat{Z}$ and the next state by CDE, and (3) cluster $\hat{Z}$s with similar direct causal effects by relational learning. CDE has the potential to deal with unsupervised environment clustering problems and so may inspire more subsequent developments in the RL community.
>
> c)	We conduct extensive experiments on mujoco and OpenAI gym to demonstrate that our method (without using either true environment ids or adaptation) achieves top-level performance by comparing with TMCL without adaptation and CaDM, and even performs slightly better than TMCL with adaptation (cf. adaptation helps identify the true environment ids).
>
> **3. In the PCA visualisations, how are the different dots of the same colour generated? Is from different subsequences of transitions from a single episode, or multiple episodes, or both?**
>
> A: Thanks. We sample and collect the context from one single episode.
>
> **4. The relational encoder take in a fixed number of past transitions, is that correct? How does this work for time-steps near the beginning of the episode, where not enough transitions are available?**
>
> A: Thanks. This is an interesting yet challenging question, which will be explored in the near future. In our paper, we follow CaDM and TMCL by using 10 past transitions as the input of the context encoder. According to the results of CaDM, 10 past transitions can achieve better performance than fewer past transitions.
> Following CaDM, we pad state-actions with zero at the beginning of the episode.
>
> **5. Have you investigated using a recurrent encoder which processes all the information in the episode?**
>
> A:  In the original paper, we use RNN-like meta-learning-based baseline, ReBAL [3]. According to Table 1, the performance of ReBAL is worse than CaDM/TMCL/Ours. This is because there is no proper constraint on the context from ReBAL.
>
> **6.	What's L_dist in the loss function in Algorithm 1?**
>
> A: L_dist is the controllable direct effect (cf. Equation 5 in the original paper) between $Z$s from the same trajectory. Because $Z$s from the same environment surely have the same controllable direct effect and $Z$s from the same trajectory belong to the same environments, we reduce the controllable direct effect between $Z$s from the same trajectory (cf. the last second paragraph in the original paper, and we have emphasized it in the revised paper).
>
>
> **7.	AugWM missing from related work: https://arxiv.org/abs/2104.05632, and seems relevant**
>
> A: Thanks for your recommendation. We have added the discussion about AugWM in Related Work of the revised paper.
>
> **8.	It would be interesting to visualise clustering of masses which are in testing distribution as well, to see how well the relational embedding generalises.**
>
> A: Thanks for your suggestion. We have added the additional visualization results as shown in Figure 13-16 in the revised paper.

---

> > ### Author Response · Authors · 2021-11-20
> > **Continue: Response to Reviewer 3e12**
> >
> > **9. The justification that their method improves through better clustering is only supported qualitatively through visualisation. It would be great to see a quantitative measure of how good the clustering is in the latent space for different methods**
> >
> > A: Thanks for your suggestion, we refer to https://scikit-learn.org/stable/auto_examples/cluster/plot_kmeans_digits.html#sphx-glr-auto-examples-cluster-plot-kmeans-digits-py and https://scikit-learn.org/stable/modules/clustering.html#clustering-evaluation to quantitatively evaluate the performance of clusters using K-means, the sample results are given below, and the whole results are given at Table 7-11 in the revised paper. Specifically, TMCL has lower clustering performances than CaDM, but TMCL still has higher returns on test environments than CaDM. This is because TMCL clusters environments via multiplying dynamics functions rather than separating Zs.
> >
> >
> > Pendulum:
> >
> > |                        | homo | compl | v-meas |  ARI  |  AMI  |
> > |:----------------------:|:----:|:-----:|:------:|:-----:|:-----:|
> > |          CaDM          |   1  | 0.655 |  0.627 | 0.516 | 0.599 |
> > |          TMCL          |   0  | 0.298 |  0.217 | 0.088 | 0.165 |
> > | Ours (no Intervention) |   0  | 0.768 |  0.762 | 0.760 | 0.653 |
> > |          Ours          |   **1**  | **0.932** |  **0.932** | **0.937** | **0.931** |
> >
> > HalfCheetah:
> >
> > |                        | homo | compl | v-meas |  ARI  |    AMI    |
> > |:----------------------:|:----:|:-----:|:------:|:-----:|:---------:|
> > |          CaDM          |   0  | 0.262 |  0.260 | 0.203 |   0.257   |
> > |          TMCL          |   0  | 0.239 |  0.165 | 0.051 |   0.126   |
> > | Ours (no Intervention) |   0  | 0.368 |  0.362 | 0.265 |   0.353   |
> > |          Ours          |   0  | **0.416** |  **0.411** | **0.312** | **0.405** |
> >
> > Cripple\_Halfcheetah:
> >
> > |      | homo | compl | v-meas |  ARI  |  AMI  |
> > |:----:|:----:|:-----:|:------:|:-----:|:-----:|
> > | CaDM |   1  | 0.733 |  0.716 | 0.686 | 0.701 |
> > | TMCL |   0  | 0.253 |  0.000 | 0.000 | 0.000 |
> > | Ours |   **1**  | **0.853** |  **0.851** | **0.860** | **0.849** |
> >
> > According to the quantitively cluster comparison (more can be found at Table 7-11 ), we can see that the cluster performance of our method outperforms baselines by a large margin, and the results are consistent with the performance on the test environments.
> >
> >
> > **10. It would be good to see a plot of average prediction error on test environment during training, to see how that changes as the methods train.**
> >
> > A: Thanks. Following your suggestion, we plot the prediction errors of test environments. Specifically, we test each test environment 10 times, and plot the average prediction error to reduce random errors. Given limited time in the rebuttal phase, we only plot the Pendulum’s and Hopper’s prediction errors on Mujoco environments as shown in Figure X in the revised paper. We can see that our approach consistently achieves lower prediction errors than baselines, and thus the claim of learning environment-separable Zs is beneficial for the generalization of dynamics functions still holds. More Mujoco environments will be validated in the final version.
> >
> >
> > **11.	You provide some intuition as to why the ablation with no intervention has lower prediction error but lower performance in 4.3, but you could test this explicitly by plotting the test prediction error and seeing whether it's higher for the no-intervention ablation**
> >
> > A:  Thanks. According to your suggestion, we plot the prediction errors on test environments in Figure 11 in the revised paper, where the model without the intervention module has higher test prediction errors than the model with intervention. The results show that the model without intervention module may over-fit on the training environments, and thus intervention module can strengthen the generalization ability.
> >
> > **12.	PCA is quite a limited visualisation technique for clustering - have you considered other techniques such as t-SNE or UMAP?**
> >
> > A: Thanks. Following your suggestion, we visualize the contexts on Pendulum and Halfcheetah using t-SNE as shown in Figure 13 and 14 in the revised paper. The visualization results show that contexts from our method are better clustered than those from either CaDM or TMCL.
> >
> > [1] Seo, Y., Lee, K., Gilaberte, I. C., Kurutach, T., Shin, J., & Abbeel, P. (2020, January). Trajectory-wise Multiple Choice Learning for Dynamics Generalization in Reinforcement Learning. In NeurIPS.
> >
> > [2] Lee, K., Seo, Y., Lee, S., Lee, H., & Shin, J. (2020, November). Context-aware dynamics model for generalization in model-based reinforcement learning. In International Conference on Machine Learning (pp. 5757-5766). PMLR.
> >
> > [3] Nagabandi, A., Finn, C., & Levine, S. (2018, September). Deep Online Learning Via Meta-Learning: Continual Adaptation for Model-Based RL. In International Conference on Learning Representations.

---

> > > ### Comment · Reviewer_3e12 · 2021-11-22
> > > **Thanks for detailed response**
> > >
> > > Thanks for the detailed response and updating the paper, I believe that will improve the clarity and robustness of your contribution. I don't plan to raise my score from an 8, but I will raise my confidence 1 point, as I'm more confident (given the additional experiments) that the contribution is meaningfully improving on prior work.

---

### Official Review · Reviewer_db3k · 2021-11-02

**Correctness:** 4
**Technical Novelty And Significance:** 3
**Empirical Novelty And Significance:** 3
**Recommendation:** 8
**Confidence:** 3

**Main Review:**

#### **Strengths**
- The paper tackles a very relevant and important problem in the RL community, where generalization to changing dynamics is one of the key aspects to enable RL algorithms for real-world applications.
- To the best of my knowledge, the novelty of the proposed method is significant. While other works have used similar ideas to characterize the environment dynamics using a vector $Z$, the paper presents an interesting idea to reduce the redundant information in $Z$, or in other words, learn a better representation of $Z$. The interventional prediction, while being simple, is elegant and works well in practice. I believe this interventional approach would open some new directions to improve current unsupervised representation learning methods, which use instance-based discrimination objectives (similar to (2) in the paper).
- The improvements of the proposed method are significant compared to the existing baselines. Visualizations such as Figures 1,7,10-13 empirical show that the proposed method actually learns semantical information of $Z$.
- The paper is well written and organized.

#### **Questions**
- What happens to the representation of $Z$ when we omit the interventional module? I would love to see a visualization similar to Figure 1. My wild guess is $Z$ will be spread out?

**Summary Of The Paper:**

This paper proposes a new method to tackle the problem of generalization to unseen dynamics for model-based RL. Previous methods learn to predict a vector $Z$ that characterizes a particular environment dynamics from past transitions. However, as the environment id or label is not available, this vector inevitably contains redundant information, which might hurt the generalization of the model. The paper therefore proposes an interventional approach to estimate the probability that two vectors $\hat{z}_i$ and $\hat{z}_j$ belong to the same environment, and then uses a relational head to force similarity between them. As a result, the redundancy in $Z$ is reduced which leads to improved generalization.

**Summary Of The Review:**

Overall, I like the idea of the paper and believe it has significant contributions to the RL community.

---

> ### Author Response · Authors · 2021-11-20
> **Response to Reviewer db3k**
>
> Thanks for your encouraging comments. We believe that the application of controllable direct effect proposed in this paper will inspire subsequent developments in the RL community. According to your suggestion, we visualize Zs learned by the model without intervention module as shown in Figure 13, 14, 15 and 16 in the revised paper, and the visualization results show that different environments’ Zs learned by the model without intervention module cannot be separated. This is because the clustering with trajectory label cannot eliminate the irrelevant information from state-action pairs, and thus contexts from different environments but similar state-actions may tend to entangle with each other.
> The supplementary experiments fully validate the effectiveness of our intervention module. If you have further concerns, we are happy to follow up.

---

### Official Review · Reviewer_q6e8 · 2021-11-02

**Correctness:** 2
**Technical Novelty And Significance:** 3
**Empirical Novelty And Significance:** 2
**Recommendation:** 6
**Confidence:** 4

**Main Review:**

Strengths
- novel idea for using CDE for clustering trajectories
- I like the separation of the two objectives (1) and (6). (1) enforces that the z's are useful for predicting dynamics. (6) enforces that the z's within the same trajectory _and_ the same environment are clustered together. The authors show that in practice the z's do end up being predictive of the environment id.
- Figure 1 shows that the learned context vectors are clearly distinguishable

Weaknesses
- the results in Figures 4 and 5 show that TMCL very similarly to the proposed method, which seems to slightly refute the authors claim that "However, because environments are not labelled, the extracted information inevitably contains redundant information unrelated to the dynamics in transition segments and thus fails to maintain a crucial property of Z: Z should be similar in the same environment and dissimilar in different ones. As a result, the learned dynamics prediction function will deviate from the true one, which undermines the generalization ability." Essentially, why is it necessary to "explicitly learn the meaningful dynamics change information" instead of doing it implicitly? The empirical results do not seem to suggest that being explicit is necessarily much more helpful for generalization and performance.
- I was interested to see that in Figure 6, not having the intervention does not seem to hurt performance or generalization that much, and in some cases even improves it. Would the authors be able to conduct an analysis similar to Figure 7 for the case where the interventional prediction is not used and test whether the contexts can still be predictive of the environment?

**Summary Of The Paper:**

This paper considers the unsupervised dynamics generalization problem. In this problem there are a set of train MDPs and a set of test MDPs, all with the same state and action spaces, but with different dynamics functions. The authors build an approaches that use past transition segments to estimate an environment-specific vector $Z$, which is then intended to act as a contextual input to the dynamics function. The problem the authors identify is that a naive application of inferring $Z$ from past transition segments undermines the ability for Z to be similar in the same environment and dissimilar in different environments. The authors claim that generalization ability is hurt by the inability to preserve this property. The authors propose a loss (equation 6) that pull $Z$'s from the same trajectory together and push away $Z$'s from different trajectories together. However, to avoid not pushing away different trajectories from the same environment, the authors propose to estimate the controlled direct effect (CDE) of the $Z$ on the next state $S_{t+1}$: if $z^i$ and $z^j$ from trajectories $i$ and $j$ have a similar CDE, then the authors declare $i$ and $j$ to be likely from the same environment. This likelihood is used as a weighting coefficient on the loss: the authors use this weighting coefficient as a "soft" indicator variable to distinguish examples from the same environment (positive examples) or examples from different environment (negative examples). Figure 1 shows that a PCA visualization of the learned context vectors using the proposed method separate the context vectors into distinct clusters. Empirical evaluation on standard benchmarks shows that the proposed method does slightly better than the next best baseline, TMCL from (https://arxiv.org/abs/2010.13303).

**Summary Of The Review:**

It's not clear how much the interventional prediction aspect of this work actually helps (see Figure 6), even though this seems to be the main proposed contribution. It is also not clear what advantage that the proposed method has over TMCL, other than the ability to qualitatively visualize distinct cluster's in the z's. But if this does not hurt performance or generalization (Fig 4 and 5), what would be the reason that the interventional prediction is crucial?

== After Rebuttal ==
The quantitative results seem to still indicate a marginal improvement over the baselines, but perhaps what is really required is to identify environments that the community cares about that would more drastically distinguish the proposed method from those that do not use the proposed intervention prediction method. However, given the current experiments, the authors have adequately addressed my concerns and therefore I have raised my score to a 6.

---

> ### Author Response · Authors · 2021-11-20
> **Response to Reviewer q6e8**
>
> Thanks for your constructive comments, and we provide specific responses and clarifications to your major concerns about our motivation and significance. After addressing these concerns, we humbly think that our contributions are clear and valuable, and thus sincerely hope that you can reconsider the decision. If you have further concerns, we are happy to follow up.
>
> **1.“Why TMCL [1] achieves better performance than CaDM [2] even if it cannot separate Zs from different environments, and TMCL’s performance is very similarly to the proposed method**
>
> A: The reason why TMCL cannot separate environment-specifics Zs is that TMCL clusters environments via multi-choice learning rather than separating Zs. Specifically, TMCL trains multiply dynamics functions, and each dynamics function in TMCL can specify a certain range of environments, and thus different dynamics functions in TMCL would have different outputs even for the same input. Therefore, TMCL does not need to separate environment-specific Zs. However, clustering environments via multiply dynamics functions rather than environment-separable Zs may raise the following issues:
>
>    a) TMCL needs an adaptation process. Our paper aims to learn a dynamics function that can generalize to zero-shot unseen environments (cf. the last line of the abstract, and the last two lines of  the introduction). However, when applying TMCL to unseen test environments, TMCL needs to choose a proper dynamics function from trained multiply dynamics functions via an adaptive planning operation (cf. the fourth paragraph of the introduction in the original submission). Therefore, it is unfair to compare the performance of our method with TMCL because our method does not conduct any adaptation process for the zero-shot deployment, but our method still achieves better performance than TMCL.
>
> For a fair comparison, we provide the average returns of TMCL without adaptation on the unseen environments in Figure 7 of the revised paper and Table given below:
>
> |                |       CaDM       |  TMCL (no adaptation)  |         Ours        |
> |:--------------:|:----------------:|:----------------:|:-------------------:|
> |    Pendulum    | -713.95$\pm$21.1 |  -726.3$\pm$23.4 |   **-587.5$\pm$64.4**  |
> |     Hopper     |  845.2$\pm$20.41 |  980.21$\pm$28.8 |  **1057.4$\pm$37.2**  |
> |   HalfCheetah  | 5876.6$\pm$799.0 | 5730.6$\pm$589.2 | **10859.2$\pm$465.1** |
> | Slim\_Humanoid |  859.1$\pm$24.01 |  1968.7$\pm$68.2 |  **2432.6$\pm$465.1** |
>
> We can see that the average returns of TMCL without adaptation (average outputs from all functions in their paper) are significantly lower than ours, especially for classic control task Halfcheetah, which is a direct evidence to support our claim: environment-separated Zs is important for the generalization of dynamics functions, and our method can significantly outperform baselines in zero-shot unseen test environments with different dynamics. Specifically, the performance of TMCL with no adaptation is still superior to the CaDM, this is because TMCL uses the invariance of Z within a trajectory (Z can predict other states within a trajectory), which is similar to our paper with no intervention module.
>
>   b) TMCL violates the data generation process, which may undermine its generalization ability. Regarding unsupervised dynamics generalization [1][2], the dynamics functions for the training and test environments are assumed to be generated from the same generalized dynamics prediction function $f$. $f$ takes environment-specified factor $Z$, state s_t, and action a_t as inputs, and outputs the next state s_{t+1}. This means, the difference between transition dynamics of different environments is only caused by the environmental specified factor z. Therefore, learning a semantically meaningful z is important for learning such generalized dynamics prediction function f, and we can extrapolate to unseen environments with the same generalized dynamics prediction function f. However, TMCL neglects the semantic information from z, and thus the learned dynamics prediction function will deviate from the true one, undermining its generalization ability. Therefore, our method outperforms TMCL without the adaptation process.
>
> **2.“Essentially, why is it necessary to "explicitly learn the meaningful dynamics change information" instead of doing it implicitly?**
>
> A: Thanks. According to the analysis above, learning meaningful environment-specified information is a key step for learning the true generalized dynamics function, and thus we can generalize well on unseen environments without adaptation. (cf. the third paragraph of Introduction in the original paper)

---

> > ### Author Response · Authors · 2021-11-20
> > **Continue: Response to Reviewer q6e8**
> >
> > **3. I was interested to see that in Figure 6, not having the intervention does not seem to hurt performance or generalization that much, and in some cases even improves it.**
> >
> > A: Thanks. Figure 6 in the main paper shows that the prediction error in the training environments, and the average returns in the test environments. We can see that the models with the intervention module have higher average returns in the test environments over the model without intervention, which is a strong yet direct evidence to support the claim “intervention module trained with controllable direct effect (CDE) can improve the generalization ability of dynamics functions”.
> >
> > Regarding prediction error, the model without intervention actually has a smaller training prediction error in the Pendulum, but it has higher prediction errors on test environments than the model with intervention (cf. Figure 11 in the revised paper). The results show that the model without intervention module may over-fit on the training environments, and thus the intervention module can strengthen the generalization ability. This phenomenon validates the value of our intervention prediction on improving the generalization ability of dynamics function. (cf. the analysis in the last three lines of section 4.3, Figure 6 and Figure 11).
> >
> > **4. Would the authors be able to conduct an analysis similar to Figure 7 for the case where the interventional prediction is not used and test whether the contexts can still be predictive of the environment?**
> >
> > A: Thanks for your suggestion. We test the similarity of contexts estimated by the model without the interventional module according to your suggestion. We use the same context similarity estimation method as that in Appendix A.7, and add the similarity statistics results in A.7
> >
> > The results of the model without intervention module show that many contexts from different environments still have high similarities. This indicates that the existing relational learning cannot separate environment-specified factors Zs. This is because clustering with only trajectory labels may mis-cluster some contexts.
> > By contrast, after incorporating the intervention module, the contexts from different environments have significantly smaller similarities than those from the same environments. The comparison between Figures 8 and 9 directly shows that our intervention module is valuable to predict whether two contexts are from the same environment or not.
> >
> > In summary, our paper aims to learn a dynamics function that can generalize on unseen environments without adaptation, while TMCL needs adaptation, our method is still better than TMCL. The supplementary results on the fair comparison with TMCL without adaptation clearly show that our method is superior to other baselines without adaptation. In addition, the key challenge of learning a generalized dynamics function is learning a semantically meaningful environment-specified factor from the historical transition segments. Therefore, we creatively introduce the controllable direct effect to identify the similarities between different learned environment-specified contexts. Last, the supplementary weight statistics (Figures 8 and 9 in the revised paper) fully validate that our intervention module can separate environment-specified Zs in an unsupervised manner. This observation may inspire subsequent developments in the RL community.
> >
> > We humbly think that we have duly addressed all your concerns, and thus sincerely hope that you can reconsider the decision. If you have further concerns, we are happy to follow up.
> >
> > [1] Seo, Y., Lee, K., Gilaberte, I. C., Kurutach, T., Shin, J., & Abbeel, P. (2020, January). Trajectory-wise Multiple Choice Learning for Dynamics Generalization in Reinforcement Learning. In NeurIPS.
> >
> > [2] Lee, K., Seo, Y., Lee, S., Lee, H., & Shin, J. (2020, November). Context-aware dynamics model for generalization in model-based reinforcement learning. In International Conference on Machine Learning (pp. 5757-5766). PMLR.

---

> > > ### Comment · Reviewer_q6e8 · 2021-11-30
> > > **Edited review**
> > >
> > > Dear Authors,
> > >
> > > Thank you for addressing my concerns. I have raised my score to a 6.

---

### Official Review · Reviewer_RE3u · 2021-11-03

**Correctness:** 3
**Technical Novelty And Significance:** 3
**Empirical Novelty And Significance:** 3
**Recommendation:** 6
**Confidence:** 4

**Main Review:**

Novelty

To the best of the reviewer’s knowledge, the paper is fairly novel. The authors use CaDM and TMCL, other model-based approaches that aim at generalization, as baselines. Figures 10-13 suggest that TMCL fails to learn separable encodings Z, while Figure 9 suggests that CaDM achieves lower returns. The proposed approach combines the best of both worlds achieving comparable returns and learning Zs that are clustered for the same domain.

Significance

The reviewer is unsure about the effect size of the proposed method. First of all, the experiments were repeated using 3 random seeds only. Combined with the overlapping confidence intervals of returns on a fair amount of test environments (e.g. Fig 5), the significance of the results is unclear. Increasing the number of runs would improve the credibility of the conclusions. For a discussion about the importance of rigorous experimentation in RL, see [1].

On a positive side, the paper follows the setup from previous papers and performs an ablation study of the proposed modification. The reviewer also appreciates qualitative findings about separability of learned Z in Figures 1 and 7 and quantitative evaluations of model prediction errors in training environments in Figures 4 and 8.

Detailed comments and questions

- Figure 1 demonstrates the linear projection of context vectors Z obtained by different methods and serves (at least partially) as a motivation for the paper. Would the conclusion that Zs for TMCL are not separable hold for a non-linear method for visualization (say, t-SNE)?
- In Table 3, C_HalfCheetah row, CaDM has a lower error but the number for the proposed method is bold.
- Conditioning $p(s’|s,a)$ on $Z_{t-k:t-1}$ makes the dynamics non-markovian. Would it be possible to use an RNN-like baseline in this setting?
- Could authors present the plots for prediction errors on *testing* environments?
- What would be a relevant model-free baseline in this setting?

[1] Henderson, Peter, Riashat Islam, Philip Bachman, Joelle Pineau, Doina Precup, and David Meger. "Deep reinforcement learning that matters." In Proceedings of the AAAI conference on artificial intelligence, vol. 32, no. 1. 2018.

-------------

POST-REBUTTAL UPDATE:

I update the score to 6 and confidence to 4

**Summary Of The Paper:**

The paper studies multi-task generalization in model-based reinforcement learning (MBRL). One of the standard ways to approach the problem is given by inferring a latent variable Z encoding each task and then conditioning the dynamics model on it. The authors propose to encode segments of state-action trajectories into Z vectors and maximize similarity between Zs from the same trajectory. Using the mechanism, the paper then studies the ability of the agents to generalize to unseen variations of the training environments. For example, if the agent was trained on MuJoCo HalfCheetah with body mass 1, it is asked to generalize to body masses 0.5 and 1.5. The paper compares the method against the baselines across 3 axes: in terms of dynamics prediction error, in terms of returns on testing environments, and in terms of separability of the inferred Z for different environments.

**Summary Of The Review:**

The reviewer leans towards recommending the submission for rejection. The ultimate objective of an RL system is maximizing the returns and it is unclear from the empirical evaluation to which extent the proposed method improves the performance of the agent on unseen tasks. However, the reviewer has limited familiarity with the related work on causality and it is possible that they missed something. Addressing the outlined concerns might improve the overall score.

---

> ### Author Response · Authors · 2021-11-20
> **Response to Reviewer RE3u**
>
> Thanks for your constructive comments. We provide below specific responses and clarifications to your concerns about the significance and experimental design of our method, and we will take all comments including those minor ones in the final version. After addressing these concerns, we humbly think that our contributions are clear and valuable, and thus sincerely hope that you can reconsider the decision. If you have further concerns, we are happy to follow up.
>
>
> **1.“Why TMCL [1] achieves better performance than CaDM [2] even if it cannot separate Zs from different environments”**
>
> A: The reason why TMCL cannot separate environment-specifics Zs is that TMCL clusters environments via multi-choice learning rather than separating Zs. Specifically, TMCL trains multiply dynamics functions, and each dynamics function in TMCL can specify a certain range of environments, and thus different dynamics functions in TMCL would have different outputs even for the same input. Therefore, TMCL does not need to separate environment-specific Zs. However, clustering environments via multiply dynamics functions rather than environment-separable Zs may raise the following issues:
>
>    a) TMCL needs an adaptation process. Our paper aims to learn a dynamics function that can generalize to zero-shot unseen environments (cf. the last line of the abstract, and the last two lines of  the introduction). However, when applying TMCL to unseen test environments, TMCL needs to choose a proper dynamics function from trained multiply dynamics functions via an adaptive planning operation (cf. the fourth paragraph of the introduction in the original submission). Therefore, it is unfair to compare the performance of our method with TMCL because our method does not conduct any adaptation process for the zero-shot deployment, but our method still achieves better performance than TMCL.
>
> For a fair comparison, we provide the average returns of TMCL without adaptation on the unseen environments in Figure 7 of the revised paper and Table given below:
>
> |                |       CaDM       |  TMCL (no adaptation)  |         Ours        |
> |:--------------:|:----------------:|:----------------:|:-------------------:|
> |    Pendulum    | -713.95$\pm$21.1 |  -726.3$\pm$23.4 |   **-587.5$\pm$64.4**  |
> |     Hopper     |  845.2$\pm$20.41 |  980.21$\pm$28.8 |  **1057.4$\pm$37.2**  |
> |   HalfCheetah  | 5876.6$\pm$799.0 | 5730.6$\pm$589.2 | **10859.2$\pm$465.1** |
> | Slim\_Humanoid |  859.1$\pm$24.01 |  1968.7$\pm$68.2 |  **2432.6$\pm$465.1** |
>
> We can see that the average returns of TMCL without adaptation (average outputs from all functions in their paper) are significantly lower than ours, especially for classic control task Halfcheetah, which is a direct evidence to support our claim: environment-separated Zs is important for the generalization of dynamics functions, and our method can significantly outperform baselines in zero-shot unseen test environments with different dynamics. Specifically, the performance of TMCL with no adaptation is still superior to the CaDM, this is because TMCL uses the invariance of Z within a trajectory (Z can predict other states within a trajectory), which is similar to our paper with no intervention module.
>
>   b) TMCL violates the data generation process, which may undermine its generalization ability. Regarding unsupervised dynamics generalization [1][2], the dynamics functions for the training and test environments are assumed to be generated from the same generalized dynamics prediction function $f$. $f$ takes environment-specified factor $Z$, state s_t, and action a_t as inputs, and outputs the next state s_{t+1}. This means, the difference between transition dynamics of different environments is only caused by the environmental specified factor z. Therefore, learning a semantically meaningful z is important for learning such generalized dynamics prediction function f, and we can extrapolate to unseen environments with the same generalized dynamics prediction function f. However, TMCL neglects the semantic information from z, and thus the learned dynamics prediction function will deviate from the true one, undermining its generalization ability. Therefore, our method outperforms TMCL without the adaptation process.
>
>
> **2.	“Would the conclusion that Zs for TMCL are not separable hold for a non-linear method for visualization (say, t-SNE)?”**
>
> A: Thanks for your suggestion. We provide the t-SNE visualization of Zs of Pendulum and Halfcheetah as shown in Figure 13 and 14 in the revised paper. We observe that Zs from TMCL are not separable, but Zs from our method are separable. The results are consistent with our analysis above: TMCL does not need separable Zs because it clusters environments via multiplying dynamics functions, but it needs adaptation when deploying it to the test data.

---

> > ### Author Response · Authors · 2021-11-20
> > **Continue: Response to Reviewer RE3u**
> >
> > **3.	“Would it be possible to use an RNN-like baseline in this setting?**
> >
> > A: In the original paper, we use RNN-like meta-learning-based baseline, ReBAL [3]. According to Table 1, the performance of ReBAL is worse than CaDM/TMCL/Ours.
> >
> > **4.	In Table 3, C_HalfCheetah row, CaDM has a lower error but the number for the proposed method is bold.**
> >
> > A: Thanks for finding this error. The original Table 3 is the prediction errors on a training environment, we have updated them with prediction errors on test environments.
> >
> > **5.	Could authors present the plots for prediction error   on testing environments?**
> >
> > A: Thanks. Following your suggestion, we plot the prediction errors of test environments. Specifically, we test each test environment 10 times and plot the average prediction error to reduce random errors. Given limited time in the rebuttal phase, we only plot the Pendulum’s and Hopper’s prediction errors on Mujoco environments as shown in Figure 11 in the revised paper. We can see that our approach consistently achieves lower prediction errors than baselines, and thus the claim of learning environment-separable Zs is beneficial for the generalization of dynamics functions still holds. More prediction errors on Mujoco environments will be validated in the final version.
> >
> > **6.	“The reviewer is unsure about the effect size of the proposed method. First of all, the experiments were repeated using 3 random seeds only.**
> >
> > A: Thanks for your comments. We agree that running more experiments would improve the credibility of the conclusions. In the original submission, we run 10 times method to reduce random errors (cf. the third line of section 4.2.2). In addition, Table 1 shows that our method can achieve at least 5.8\% improvement over baselines, so the superiority of our method over baselines is significant.
> >
> > In the revised version, we further emphasized that we run our method 10 times to reduce random errors. “We run each test environment 10 times and calculate the mean of returns of all test environments as the final test returns, and so the empirical comparison is reliable.” Moreover, we also run more Mujoco environments on Swimmer and Slim_Humanoid_Standup to show the superiority of our method. Specifically, their environmental settings are the same as the setting of Slim_Humanoid.
> >
> > | | Cadm [2]              | TMCL [1]           | Ours               |
> > |-----------------------|--------------------|--------------------|--------------------|
> > | Slim_Humanoid_StandUp | 1221.3 $\pm$ 337.7 | 1450.6 $\pm$ 182.5 |  **1651.5 $\pm$ 201.7** |
> > | Swimmer               | 28.8 $\pm$ 2.1     | 32.58 $\pm$1.66   | **37.2 $\pm$ 1.8**   |
> >
> > We can that the performance of our method is higher than baselines by 13.8% and 14.1% for Slim_Humanoid_StandUp and Swimmer, respectively. The results show that the generalization ability of our method is consistently improved.
> >
> > **7.	What would be a relevant model-free baseline in this setting?**
> >
> > A: CaDM provides the performance of model-free baselines e.g. Pearl, PPO. However, because model-free baselines are of low data efficiency, they cannot perform as well as model-based methods given the same amount of data (2 * 10 ^5).
> >
> > **8.	The reviewer has limited familiarity with the related work on causality and it is possible that they missed something.**
> >
> > A: To the best of our knowledge, this paper is a pioneer to use controllable causal effect to estimate the environment ID in an unsupervised manner. Figure 8 in the revised submission shows that our method can estimate the similarity of contexts from unknown environments. It thus can inspire subsequent works in the RL community.
> >
> > In summary, for the zero-shot dynamics generalization (a more practical setting of the unsupervised dynamics generalization [1][2]), our method outperforms TMCL without adaptation as shown by the supplementary experimental results and the new validations provided in this rebuttal and still slightly outperforms TMCL as shown in the main text. However, it is not fair to compare our method with TMCL, because TMCL uses an adaptation process while our method does not, let alone the adaptation is not allowed for the zero-shot dynamics generalization.
> >
> > We humbly think that we have duly addressed all your concerns, and thus sincerely hope that you can reconsider the decision. If you have further concerns, we are happy to follow up.
> >
> >
> > [1] Seo, Y., Lee, K., Gilaberte, I. C., Kurutach, T., Shin, J., & Abbeel, P. (2020, January). Trajectory-wise Multiple Choice Learning for Dynamics Generalization in Reinforcement Learning. In NeurIPS.
> >
> > [2] Lee, K., Seo, Y., Lee, S., Lee, H., & Shin, J. (2020, November). Context-aware dynamics model for generalization in model-based reinforcement learning. In I (pp. 5757-5766). PMLR.
> >
> > [3] Nagabandi, A., Finn, C., & Levine, S. (2018, September). Deep Online Learning Via Meta-Learning: Continual Adaptation for Model-Based RL. In ICLR

---

> > > ### Comment · Reviewer_RE3u · 2021-11-23
> > > **Response to Authors**
> > >
> > > Dear Authors,
> > >
> > > Thank you for the reply and detailed comments.
> > >
> > > I believe you have addressed my major concerns, thus I increase the score to 6 and the confidence to 4.

---

### Author Response · Authors · 2021-11-20
**To All Reviewer**

We appreciate all the reviewers for their valuable and constructive comments. We have revised our paper (with blue color) accordingly and summarized the main improvements below:
1.	Reviewer RE3u and q6e8 mention that TMCL does not have the separated environment-specified Zs but have good performance. This is because TMCL clusters environments via multi-choice learning rather than separating Zs. However, TMCL needs adaptation process during inference while our method does not. In order to obtain the fair comparison, we add the results of TMCL without adaptation in the Figure 7 of the revised paper. The results can resolve the significance concern from Reviewer RE3u and q6e8.
2.	We plot the prediction errors on the test environments at Figure 7 following the suggestion of Reviewer RE3u and 3e12
3.	We add the T-SNE visualization results of learned contexts Z at Figure 13 and 14 following the suggestion of Reviewer RE3u and 3e12.
4.	We add the weight statistics visualization as Figure 8 and 9 in the revised paper following the suggestion of Review q6e8
5.	We add the quantitative evaluation results of clusters as Table 7-11  in the revised paper following Reviewer 3e12.
6.	We add the contexts visualization of model without intervention at Figure 13-16 following the suggestion of Reviewer db3k.

In addition, we provide specific response to each reviewer and duly addressed all concerns, and thus sincerely hope that Reviewer RE3u and q6e8 can reconsider the decision.

---

### Decision · Program_Chairs · 2022-01-20

**Decision:**

Accept (Poster)

**Comment:**

Description of paper content:

The authors propose a dynamics model that can generalize to novel environments. The train and test MDPs have the same state and action spaces but different dynamics. Environment specific inference is achieved by estimating latent vectors Z that describe the non-stationary or variable part of the dynamics. These Z-s are inferred from trajectory segments in unlabeled environments. The Z-s are learned contrastively: Z-s from the same trajectory are pulled together, and Z-s from separate trajectories are pushed apart. However, to mitigate the error of distancing Z-s from different trajectories but the same environment, Z-s on trajectories with similar transitions are also pushed together using a soft clustering penalty. These losses are justified based on ideas from Pearl’s causal inference.

Summary of paper discussion:

The reviewers concluded that the contributions are conceptually interesting and “somewhat” novel. The reviewers felt that the empirical performance gains of the method over baselines were demonstrated but not extremely impressive.